# Learning Symbolic Rules for Reasoning in Quasi-Natural Language

**Kaiyu Yang**[*]                                                                 *kaiyuy@cs.princeton.edu*
*Department of Computer Science*
*Princeton University*

**Jia Deng**                                                                      *jiadeng@cs.princeton.edu*
*Department of Computer Science*
*Princeton University*

**Reviewed on OpenReview:** *https://openreview.net/forum?id=gwRwHUZUgz*

## Abstract

Symbolic reasoning, rule-based symbol manipulation, is a hallmark of human intelligence. However, rule-based systems have had limited success competing with learning-based systems outside formalized domains such as automated theorem proving. We hypothesize that this is due to the manual construction of rules in past attempts. In this work, we take initial steps towards rule-based systems that can reason with natural language but without manually constructing rules. We propose MetaQNL, a "Quasi-Natural Language" that can express both formal logic and natural language sentences, and MetaInduce, a learning algorithm that induces MetaQNL rules from training data consisting of questions and answers, with or without intermediate reasoning steps. In addition, we introduce soft matching—a flexible mechanism for applying rules without rigid matching, overcoming a typical source of brittleness in symbolic reasoning. Our approach achieves state-of-the-art accuracies on multiple reasoning benchmarks; it learns compact models with much less data and produces not only answers but also checkable proofs. Further, experiments on two simple real-world datasets demonstrate the possibility for our method to handle noise and ambiguity.[1]

## 1 Introduction

Symbolic reasoning—rule-based symbol manipulation—is a core component of human intelligence (Mercier & Sperber, 2017). It has also been a core part of computer science research, and has achieved significant success in domains such as software verification (Darvas et al., 2005) and theorem proving (Kovács & Voronkov, 2013). However, such success has been restricted to domains amenable to rigid, precise formalization. It remains a challenge how to translate such success into "informal" domains such as reasoning with common-sense knowledge and natural language input. Prior attempts to build rule-based systems, which rely on manually constructed rules, have achieved limited success and tended to produce brittle systems.

Deep learning provides an attractive alternative that can easily sidestep the question of representation. Deep networks can be trained to perform a reasoning task by directly predicting the answer without explicit symbol manipulation (Clark et al., 2020). However, they can require a large amount of training data and can suffer from poor generalization. More importantly, unlike symbolic systems, a deep network is a black box that is hard to interpret and verify. Such lack of interpretability is undesirable in certain applications, especially those critical to safety and security.

In this work, we ask how to build a rule-based system that reasons symbolically but can work with natural language and handle domains difficult to formalize. Such a system would perform reasoning by explicit

---

[*]Research conducted while Kaiyu Yang was at Princeton University.
[1]The code is available at `https://github.com/princeton-vl/MetaQNL.jl`.

symbol manipulation based on rules, therefore is more interpretable and verifiable, but at the same time flexible enough for natural language.

At a glance, this may appear a large departure from the conventional wisdom that learning systems, particularly deep networks, are far superior to rule-based systems, as history has demonstrated repeatedly. However, we hypothesize that this conventional wisdom is incorrect because it assumes a false dichotomy between using learning and using rules; rule-based systems underperformed not because they were rule-based, but because it is difficult to construct rules manually. Further, we hypothesize learning rules from data is key to building effective rule-based systems, but it may require a different kind of learning than gradient descent.

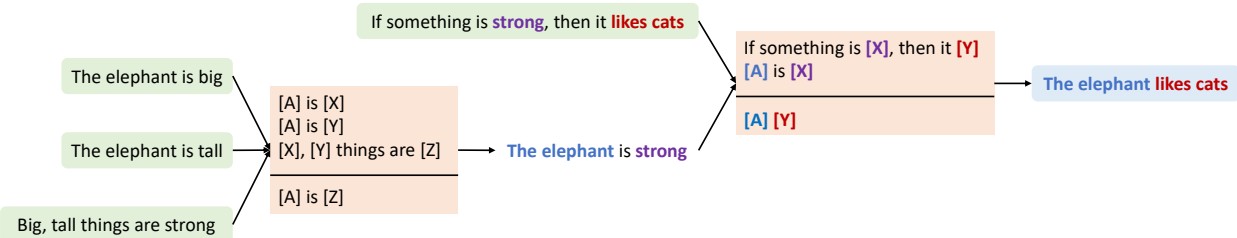

Figure 1: Example proof in MetaQNL with 4 assumptions, 1 goal, and 2 rule applications. Each rule has multiple premises and one conclusion. They can have variables ([A], [X], [Y], etc.) that bind to concrete sentences during reasoning. The resulting proof is interpretable to humans and checkable by machines. When applying MetaQNL to a specific task, assumptions and the goal are usually given, and the system has to (1) learn a set of rules from data, and (2) apply the rules to find proofs. The name "MetaQNL" is inspired by Metamath (Megill & Wheeler, 2019)—a formal language for expressing mathematical theorems/proofs, in which substitution is the only rule of inference.

Our goal is thus to develop a method that automatically learns symbolic rules from data to enable rules-based reasoning with natural language. This poses two main questions. First, what is the system of rules—the basic structures defining what symbols and manipulations are allowed—such that it is compatible with not only logic but also natural language? Second, what is the learning algorithm that induces rules from data?

In this work, we take initial steps towards answering both questions. We propose *MetaQNL*, a symbolic system we call a "Quasi-Natural Language", which is compatible with not only rigorous logical inference but also natural language (Fig. 1). We prove theoretical results showing that MetaQNL is Turing complete. In addition, we propose *MetaInduce*, a learning algorithm that induces MetaQNL rules from training data consisting of questions and answers, with or without intermediate reasoning steps. It encodes the rule learning problem as a maximum satisfiability (MAX-SAT) problem, which can be solved efficiently by existing solvers.

Reasoning in natural language can be fuzzy and ambiguous. We extend MetaQNL to support such reasoning through soft matching—relaxing the rigid matching conditions when applying rules. Soft matching is a potential avenue for applying MetaQNL to real-world, free-form natural language. And it enables integration with state-of-the-art large pretrained language models (Raffel et al., 2020).

**Overview of Results.** We benchmark our method on 3 tasks: learning compositional instructions, logical reasoning, and morphological analysis. For compositional instructions, our method not only achieves 100% accuracy on MiniSCAN (Lake et al., 2019) and SCAN (Lake & Baroni, 2018), but also recovers the ground truth rules. For logical reasoning, it achieves state-of-the-art performance on RuleTaker (Clark et al., 2020), including the noisy data paraphrased by crowd workers. For morphological analysis, it learns morphological rules from real-world linguistic data and is competitive with neural seq2seq models in some languages. Compared to existing methods, our approach learns compact models with much less data, and produces not only answers but also checkable proofs. On RuleTaker, we learn a model that has only 2869 symbols but is competitive with a prior approach that uses a neural network with 11 billion parameters (Tafjord et al., 2021). These results demonstrate the promise of MetaQNL/MetaInduce as a radically different learning approach. Finally, we discuss the limitations of our current method, as well as potential improvements in the future to make it general and scalable enough for more realistic, free-form natural language.

## 2 Related Work

**Symbolic Reasoning.** Symbolic reasoning has been studied extensively in classical AI (Robinson & Voronkov, 2001; Yang & Deng, 2019) and cognitive science (Goodman et al., 2008; Piantadosi et al., 2016). An open problem is to handle domains without a natural formalization, e.g., images or texts. One common approach is to manually construct a formal system (e.g., first-order logic with manually defined functions and predicates) and then perform semantic parsing to convert images/texts into formalized statements (Mao et al., 2018; Dai et al., 2019; Saparov & Mitchell, 2022). In contrast, our approach does not require a semantic parser, because MetaQNL rules consist of premises/conclusions that can take the form of natural language sentences.

Natural Logic (McAllester & Givan, 1993; MacCartney & Manning, 2007) is another class of symbolic systems defined using the syntax of natural language, bypassing semantic parsing. It is highly specialized, committing to predefined rules for monotonicity reasoning (Icard III & Moss, 2014). In contrast, MetaQNL has no such restrictions because it is not a specific logic but a meta-language with minimal structures such that it can instantiate various types of reasoning.

None of prior works discussed so far learn rules from data; instead, they use predefined formal systems that are already specialized and already encode a substantial amount of prior knowledge. In contrast, MetaQNL is almost "knowledge-free" in the sense that it imposes the weakest possible structure on the permitted rules and lets the specific rules emerge from data through learning.

**Reasoning with Neural Networks.** Neural networks can perform "soft" reasoning in the space of continuous vectors without manipulating discrete symbols explicitly (Sanyal et al., 2022; Dalvi et al., 2022; Sprague et al., 2022; Creswell et al., 2023). Clark et al. (2020) finetune a Transformer (Vaswani et al., 2017) to classify whether the goal is provable from the assumptions (both are sentences in natural language). Tafjord et al. (2021) and Yang et al. (2022) go one step further to generate proofs. And Bostrom et al. (2021) generate conclusions from premises.

Beyond Transformers, researchers have tried to incorporate inductive biases inspired by symbolic reasoning into neural networks, leading to *neuro-symbolic architectures.* One line of work is Neural Theorem Provers (Rocktäschel & Riedel, 2017; Weber et al., 2019), which uses backward chaining, a classical symbolic reasoning algorithm, to construct the network architecture dynamically. Another line of work embeds symbolic structures into continuous vectors while preserving logical operations as continuous operations between vectors (Grefenstette, 2013; Cohen et al., 2018; Lee et al., 2016; Schlag et al., 2019).

Unlike these works, we learn symbolic rules instead of neural network weights. Further, during inference, we generate symbolic proofs whose correctness is guaranteed and can be mechanically checked (with respect to the rules). Tafjord et al. (2021) and Yang et al. (2022) also generate proofs, but their proofs are natural language texts whose correctness is neither guaranteed nor mechanically checkable—they train neural networks to generate proofs directly but do not expose a formal system of rules against which a proof can be checked.

**Learning Rules from Data.** Inductive logic programming (ILP) learns rules in logic programs such as Datalog (Muggleton, 1991). Extending ILP to natural language is non-trivial—due to the infeasible need for a predefined ontology of objects/predicates, as well as a perfect semantic parser. Unlike ILP, rules in MetaQNL can express not only logic programs but also natural language sentences. And our experiments show that MetaQNL can solve tasks where ILP is not easily applicable.

That said, our method for learning rules draws inspiration from ILP approaches that encode proofs as SAT problems (Raghothaman et al., 2019). However, we deal with a richer and more complex space of rules. Our rules consist of sentences with variables, and they form a rich hierarchy from abstract rules to concrete ones, making the search space for rule learning much larger. In contrast, rules in ILP are more constrained. They are typically Horn clauses in first-order logic. And they impose strong syntactic constraints, e.g., using templates (Raghothaman et al., 2019), or by restricting to binary predicates (Evans & Grefenstette, 2018). These constraints are critical to good performance but are domain-dependent and difficult to get right (Cropper & Dumančić, 2022). Over-constraining the rule space makes the system less expressive, less generally applicable, and more brittle in the presence of noise. Another difference is that we minimize the number of rules in order to generalize, which is unnecessary for ILP due to stronger syntactic constraints.

Beyond ILP, rule learning has also been explored in other contexts. RNNLogic (Qu et al., 2021) uses RNNs to generate rules for knowledge base completion. They impose strong syntactic constraints that rules must be expressed as a sequence of predicates, making them less suitable for more general reasoning. Nye et al. (2020) learn rules for a string rewriting system. MetaQNL is more general because it can be applied to not only string rewriting but also other forms of reasoning (Sec. 6). Similar to us, Unification Networks (Cingillioglu & Russo, 2020) use concrete examples to learn abstract rules with variables. However, their system of rules is significantly less general than ours: their variables can only bind to a single word, whereas our variables bind to arbitrary sentence fragments. In addition, their system does not support multi-step reasoning. All reasoning is done in a single step: producing a conclusion in the form of an answer ("yes/no", a number, etc.) given premises consisting of a question and a set of supporting facts.

## 3  MetaQNL: A Symbolic System in Quasi-Natural Language

In this section, we formally define MetaQNL and present theoretical results showing that MetaQNL is Turing complete. We call MetaQNL a quasi-natural language because it has a formal syntax but is compatible with natural language. Like in natural language, a *sentence* in MetaQNL is simply a sequence of tokens. There are 3 different types of tokens—words, variables, and special symbols. Taking the sentence "`$FALSE$ The elephant likes [X]`" as an example, "`The`", "`elephant`" and "`likes`" are words. They are treated as pure symbols without any semantic meanings. "`[X]`" is a variable—a placeholder that binds to concrete sentences in reasoning. "`$FALSE$`" is a special symbol. They are useful for encoding the structures of specific tasks, which will become more clear in Sec. 6. In this paper, we delimit special symbols with \$. Sentences without variable are called *concrete sentences*, e.g., "`$FALSE$ The elephant likes cats`".

**Definition 3.1** (Sentence). Let $\Sigma_w, \Sigma_v, \Sigma_s$ be vocabularies of words, variables, and special symbols; they are disjoint and countable. Let $\Sigma = \Sigma_w \cup \Sigma_v \cup \Sigma_s$, then any $t \in \Sigma$ is a token. A sentence $s = (t_1, t_2, \ldots, t_n) \in \Sigma^+$ is a non-empty sequence of tokens. A concrete sentence is a sentence without any variable, i.e., $\forall i, t_i \notin \Sigma_v$.

MetaQNL expresses permitted reasoning steps through *rules*. A rule has multiple sentences as its premises ("`The elephant [X]`", "`If something [X] then it [Y]`") and one sentence as the conclusion ("`The elephant [Y]`"). Intuitively, the conclusion should follow from the premises regardless of what values the variables take. *Concrete rules* are rules without variables.

**Definition 3.2** (Rule). A rule takes the form of $p_1; p_2; \ldots; p_n \vdash c$, with $n$ premises $p_1, p_2, \ldots, p_n \in \Sigma^+$ and a conclusion $c \in \Sigma^+$. It is a concrete rule if all premises and the conclusion are concrete sentences.

In reasoning, rules are made concrete by substituting variables with concrete sentences. Given the rule $r_1 =$ "`The elephant [X]; If something [X] then it [Y]` $\vdash$ `The elephant [Y]`", we can instantiate it with the substitution $\{[X] \rightarrow \texttt{is strong}, [Y] \rightarrow \texttt{likes cats}\}$, deriving the concrete rule $r_2 =$ "`The elephant is strong; If something is strong then it likes cats` $\vdash$ `The elephant likes cats`". Here we say $r_1$ is more general than $r_2$, or equivalently, $r_2$ is an instance of $r_1$. Applying substitutions makes a sentence/rule more specific, introducing a partial order among sentences/rules (proofs in Appendix A).

**Definition 3.3** (Substitution). Let $\Sigma_{-s}^+ = (\Sigma_w \cup \Sigma_v)^+$ be the set of sentences with only words and variables (without special symbols). A substitution $\sigma$ is a function from $\Sigma_v$ to $\Sigma_{-s}^+$.[2] Substitutions can be extended to be functions on tokens, sentences, and rules. Given a token $t \in \Sigma$, applying the substitution $\sigma$ produces a sentence $\sigma t$, which equals to $\sigma(t)$ if $t$ is a variable. Otherwise, $\sigma t$ is just $t$ itself. Given a sentence $s = (t_1, t_2, \ldots, t_n)$, applying the substitution $\sigma$ produces $\sigma s = (\sigma t_1, \sigma t_2, \ldots, \sigma t_n)$.[3] Given a rule $r = p_1; p_2; \ldots; p_n \vdash c$, applying $\sigma$ produces another rule $\sigma r = \sigma p_1; \sigma p_2; \ldots; \sigma p_n \vdash \sigma c$.

**Definition 3.4** (Partial order among sentences and rules). Let $s_1$ and $s_2$ be two sentences, $s_2$ is an instance of $s_1$ (denoted by $s_2 \leq s_1$) if and only if there exists a substitution $\sigma$ such that $s_2 = \sigma s_1$. In this case, we also say $s_1$ is more general than $s_2$. Similarly, let $r_1$ and $r_2$ be two rules, $r_2$ is an instance of $r_1$ (denoted by $r_2 \leq r_1$) if and only if $\exists \sigma, r_2 = \sigma r_1$.

---

[2]A substitution is defined on the set of all variables, but in practice it only involves a few. We can think of it as being the identity function for other variables. This convention makes it easier to composite substitutions as function composition.

[3]We are abusing notations to treat a token and a single-token sentence interchangeably.

In reasoning (Fig. 1), the prover is given a set of rules $\mathcal{M}$, multiple concrete sentences $A$ as assumptions, and one sentence $g$ as the goal. It iteratively instantiates concrete rules from $\mathcal{M}$ and applies them to generate a proof of $g$. Similar to Prolog, $g$ may have variables ("The elephant [X]"), and the prover succeeds if it proves any instance of $g$ (e.g., "The elephant sleeps").

**Definition 3.5** (Proof). A proof $P = (V, E)$ is a directed acyclic graph whose vertices $V$ are concrete sentences or concrete rules. For each concrete rule $r = p_1; p_2; \ldots p_n \vdash c \in V$, it must satisfy two conditions: (1) $r$ connects to its conclusion $c \in V$ via an edge $(r, c) \in E$; (2) For each premise $p_i$, we have $p_i \in V$ and $(p_i, r) \in E$. Besides these edges, there is no other edge in $E$. In addition, the proof $P$ can have multiple sentences without inbound edges (assumptions) but only one sentence without outbound edges (goal).

**Definition 3.6** (Theorem proving). Given a set of rules $\mathcal{M} = \{r_1, r_2, \ldots, r_k\}$, concrete sentences $A = \{a_1, a_2, \ldots, a_n\}$ as assumptions, and a sentence $g$ as the goal, the theorem prover tries to find a proof $P$ such that: (1) $P$'s assumptions are $A$. (2) $P$'s goal is an instance of $g$. (3) Rules in $P$ are instances of rules in $\mathcal{M}$.

Now we have defined MetaQNL as a symbolic system. To characterize its theoretical expressiveness, we note that MetaQNL is Turing complete (Theorem 3.7 is proved in Appendix B). The implication is encouraging— In principle, MetaQNL can solve any task to the extent that the task is solvable by computer programs. Though in practice, challenges may arise from how to express the task in an appropriate form and how to learn a suitable set of rules.

**Theorem 3.7** (Turing completeness of MetaQNL). *For any recursively enumerable language $L$, it is possible to construct a set of MetaQNL rules for recognizing $L$.*

## 4 MetaInduce: Learning Rules from Data

### 4.1 Problem Setup and Loss Function

Learning rules is a machine learning problem where the model consists of not continuous weights but symbolic rules. The problem setup is familiar: We want to use the training set $\mathcal{D}_{\text{train}}$ to find a model that performs well not only on $\mathcal{D}_{\text{train}}$ itself but also on the test set $\mathcal{D}_{\text{test}}$. For MetaQNL specifically, the training set $\mathcal{D}_{\text{train}} = \{\mathcal{D}_{\text{train}}^+, \mathcal{D}_{\text{train}}^-\}$ consists of a set of *provable* examples $\mathcal{D}_{\text{train}}^+$ and a set of *unprovable* examples $\mathcal{D}_{\text{train}}^-$. They both contain training examples in the form of $(A_i, g_i)$, where $A_i$ is a set of assumptions and $g_i$ is the goal. Intuitively, provable examples are positive examples demonstrating sound logical inference, whereas unprovable examples are negative examples demonstrating unsound inference. More formally, a model $\mathcal{M}$ is consistent with a provable example $(A_i, g_i) \in \mathcal{D}_{\text{train}}^+$ if $g_i$ is provable from $A_i$ using rules in $\mathcal{M}$. Similarly, $\mathcal{M}$ is consistent with an unprovable example $(A_i, g_i) \in \mathcal{D}_{\text{train}}^-$ if $g_i$ cannot be proved from $A_i$.

Given only $\mathcal{D}_{\text{train}}$, we need to find a model that is consistent with as many examples in $\mathcal{D}_{\text{test}}$ as possible. However, it is not sufficient to optimize the consistency with training data, because there is a trivial model that performs perfectly in training but fails in testing—one rule per example. That is, any example $(A_i, g_i) \in \mathcal{D}_{\text{train}}^+$ with $A_i = \{a_1, a_2, \ldots, a_k\}$ is provable using the rule $a_1; a_2; \ldots; a_k \vdash g_i$. To prevent trivial solutions, we need to penalize the model complexity. Here we measure it as the number of rules and minimize a loss function that evaluates both model complexity and consistency with training data:

$$\mathcal{L}(\mathcal{M}) = |\mathcal{M}| - \lambda^+ \mathcal{N}(\mathcal{M}, \mathcal{D}_{\text{train}}^+) - \lambda^- \mathcal{N}(\mathcal{M}, \mathcal{D}_{\text{train}}^-). \tag{1}$$

$|\mathcal{M}|$ is the number of rules; $\mathcal{N}(\mathcal{M}, \mathcal{D}_{\text{train}}^+)$ and $\mathcal{N}(\mathcal{M}, \mathcal{D}_{\text{train}}^-)$ are the number of provable/unprovable examples consistent with $\mathcal{M}$. $\lambda^+$ and $\lambda^-$ are hyperparameters controlling the trade-off between the three terms.

The optimization is challenging. Even a single evaluation of $\mathcal{L}(\mathcal{M})$ is expensive: $\mathcal{N}(\mathcal{M}, \mathcal{D}_{\text{train}}^+)$ and $\mathcal{N}(\mathcal{M}, \mathcal{D}_{\text{train}}^-)$ require running the prover on all training examples. Further, it is much harder to find the optimal $\mathcal{M}$ due to the combinatorial and non-differentiable search space. We introduce MetaInduce, a general method for learning rules by encoding Eqn. 1 as a maximum satisfiability (MAX-SAT) problem, which can be solved efficiently by existing solvers.

---

**Algorithm 1:** MetaInduce

---

**Input**   : Training data $\mathcal{D}_{\text{train}} = \{(A_i, g_i)\}_{i=1}^n$; $A_i$ is the assumptions; $g_i$ is the goal.
**Output:** Model $\mathcal{M}$ consisting of a set of rules
**1** $\mathcal{M} \leftarrow \varnothing$
**2** **for** $j \leftarrow 1$ **to** *num_epochs* **do**
**3**  |  **for** $i \leftarrow 1$ **to** $n$ **do**
**4**  |  |  candidates $\leftarrow$ `propose_rules`$(\mathcal{D}_{\text{train}}, i)$
**5**  |  |  `prove`$(A_i,\ g_i,\ candidates \cup \mathcal{M})$
**6**  |  rules $\leftarrow$ `abstract_rules`()
**7**  |  $\mathcal{M} \leftarrow$ `prune_rules`(rules)

---

### 4.2  MetaInduce Algorithm

**Overview.**   MetaInduce is outlined in Algorithm 1. Similar to SGD for training neural networks, MetaInduce goes through the training data for several epochs; during an epoch, it processes one example per iteration. Given an example $(A_i, g_i)$ (either provable or unprovable), it first relies on a *rule proposer* for generating candidate rules that are concrete and potentially useful for proving $g_i$ from $A_i$. Then it runs an existing prover to search for proofs, using both the candidate rules and existing rules in the model. At the end of each epoch, MetaInduce abstracts all concrete rules used in the proofs into rules with variables. Then it performs rule pruning—selecting $\mathcal{M}$ as a subset of the rules minimizing the loss (Eqn. 1). Next, we explain each step in more detail.

**Rule Proposal.**   It is desirable to have a learning system that can learn with minimal domain knowledge but learn more efficiently when more knowledge is available. Therefore, we design the rule proposers to be *domain-dependent*, which enables incorporating domain knowledge into the system for more efficient learning. Specific designs of rule proposers used in this paper can be found in Sec. 6 and Appendix D, E. However, domain-dependent rule proposers do not significantly compromise our method's general applicability to different tasks. As argued in Sec. 6, rule proposers used in this paper only make modest task-specific assumptions—not more than the assumptions made by prior works. Also, the rule proposer has substantial room for future research (Sec. 7). One particularly exciting direction is to train deep neural networks for proposing rules, which can reduce the need for manually crafted heuristics.

Furthermore, a good rule proposer alone—if not used as a part of MetaInduce—is not sufficient for learning rules. First, the rule proposer only generates concrete rules. It is up to MetaInduce to abstract them into rules with variables. Second, the rule proposer only generates rules useful for a single training example, whereas MetaInduce learns rules useful for the entire dataset. Third, the rule proposer does not have to be accurate. MetaInduce can reliably learn correct rules even if most candidate rules are wrong (see Sec. 6.1).

**Theorem Proving.**   Theorem proving in MetaQNL is straightforward, thanks to existing algorithms such as forward/backward chaining (Russell & Norvig, 2002). Forward chaining starts with the assumptions and applies rules to derive new sentences until the goal is reached. Conversely, backward chaining starts with the goal and applies rules in the reverse direction until all assumptions are satisfied. There may be multiple proof paths, and our implementation finds all of them up to a predefined depth limit.

**Rule Abstraction.**   The proofs contain only concrete rules, and we have to generalize them into rules with variables. We use a symbolic procedure called *anti-unification* (Plotkin, 1970). Given two rules $r_1$ and $r_2$, it attempts to find the most specific rule $r$ such that $r_1 \leq r$ and $r_2 \leq r$ (analogous to the lowest common ancestor of two nodes in a tree; see Fig. 2 (*Left*) for examples and Appendix C for more details). Our algorithm performs anti-unification by recursively matching the beginning of two sentences.

Let $\Gamma$ be the set of all concrete rules in the proofs. We iteratively anti-unify rules in $\Gamma$ and add the result back, until no new rule can be generated. The result is denoted by $\Gamma'$, which contains not only concrete rules but also their generalizations.

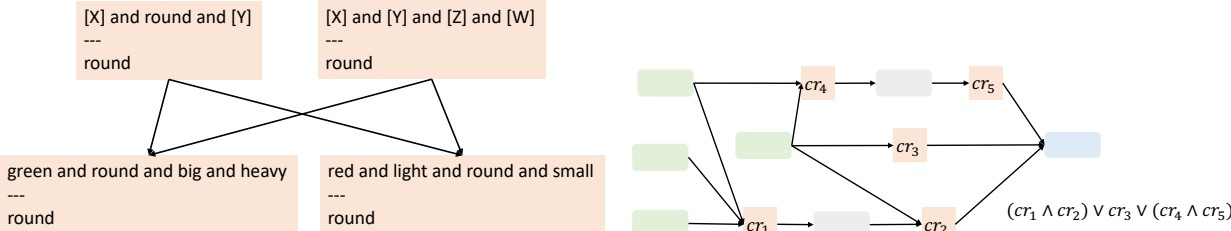

Figure 2: *Left*: Examples of anti-unifying two concrete rules into more abstract rules. Anti-unification may have multiple solutions that are not comparable with each other. *Right*: Encoding as a boolean constraint a proof with 3 paths from assumptions to the conclusion. Each concrete rule $cr_i$ corresponds to a boolean variable. The proof is a disjunction of all paths; each path is a conjunction of concrete rules.

**Rule Pruning.** Rule pruning selects $\mathcal{M}$ as a subset of $\Gamma'$ by encoding all proofs as a MAX-SAT problem, whose solution corresponds to a set of rules that approximately minimizes the loss function Eqn. 1. We encode each rule $r \in \Gamma'$ using a boolean variable (also denoted $r$). $r = 1$ means the rule should be included in $\mathcal{M}$. For any concrete rule cr $\in \Gamma'$, we have an additional boolean variable cr. cr $= 1$ means cr is necessary for proving the training examples. We impose 3 different types of constraints on these boolean variables:

- *Data consistency*: As shown in Fig. 2 (*Right*), for the $i$th training example, its proof $P_i$ may have many paths from the assumptions to the goal, but the example is provable as long as one of them is valid. For provable examples, we encode $P_i$ as a disjunction of proof paths. Each path is valid if and only if all concrete rules along the path are valid. So we encode a proof path as a conjunction of all cr boolean variables it contains. Analogously, for unprovable examples, we simply take the negation of the previous boolean formula to encourage the absence of a valid proof. Finally, a good model is not necessarily consistent with every training example. So $P_i$ is encoded as a soft constraint with weight $\lambda^+$ or $\lambda^-$.

- *Model complexity*: To minimize the number of rules, we add a soft constraint $\neg r$ of weight 1 for each $r$ boolean variable. It encourages $r = 0$.

- *Rules instantiation*: Each concrete rule cr must be an instance of a rule $r$. Let $r_1, r_2, \ldots, r_k \in \Gamma'$ be the set of all rules in $\Gamma'$ such that $cr \leq r_i$. cr can be instantiated only if at least one of them is in the model $\mathcal{M}$. Therefore, we add a hard constraint cr $\rightarrow r_1 \vee r_2 \vee \cdots \vee r_k$.

Given the boolean constraints above, a MAX-SAT solver finds an assignment of boolean variables to minimize the combined weights of violated constraints, which equals to Eqn. 1. Therefore, it learns a set of rules minimizing our loss function.

### 4.3 Computational Complexity and Scalability

Let $n$ be the number of training examples, the complexities of rule proposal and theorem proving are $O(n)$ since they are performed independently for each example. Rule abstraction is $\Omega(n^2)$ since we anti-unify all pairs of rules. However, rule pruning is NP-hard in the worst case due to the use of MAX-SAT (Krentel, 1986). Therefore, our system, in its current form, would struggle with very large $n$. In order to scale to millions of examples, future work would have to reduce the complexity to $O(n)$.

For rule abstraction, note that our pairwise anti-unification is only a design choice at the implementation level, rather than a conceptual necessity. In principle, all we need is a procedure for generating abstract rules from concrete ones. It is possible to develop more efficient algorithms for rule abstraction with linear or even sublinear complexity. They can also be based on machine learning, e.g., Cingillioglu & Russo (2020).

For rule pruning, one way to scale it up is through approximate MAX-SAT solvers, which are faster but not guaranteed to find the global optimum. Similar to deep learning, MetaInduce does not require the global optimum to work well in practice. Another direction is learning in mini-batches. It avoids encoding the entire dataset as a large MAX-SAT problem, and instead solves multiple small MAX-SAT problems.

In addition to theoretical complexity, we also report MetaInduce's experimental run time in Sec. 6, which compares favorably to existing methods based on deep neural networks. Our run time is noteworthy considering that deep neural networks are trained on software/hardware stacks highly optimized for them (GPUs, cuDNN, PyTorch, etc.), whereas MetaInduce can run even on a laptop without GPUs. Our method is at an early stage without nearly as much engineering effort for improving the run time efficiency. At this stage, it is difficult to perform a fair and comprehensive comparison of the run time (e.g., using the same hardware).

## 5 Soft Matching

Similar to classical theorem proving, reasoning in MetaQNL relies on precise and rigid matching between rules and assumptions. For example, the rule "`The [A] smiles ⊢ Someone smiles`" is not applicable to "`An elephant smiles`" due to the lack of "`The`". Although precise matching guarantees the rigor of reasoning, reasoning in natural language is often fuzzy and ambiguous, without the same degree of rigor as a mathematical proof. Supporting fuzzy reasoning is necessary for MetaQNL to cover a broader spectrum of reasoning in natural language. It requires us to relax the rigorous proofs in Definition 3.6 to fuzzy proofs with scores indicating the degree of rigor. Another benefit of soft matching is that it allows the system to degrade gracefully—it can produce an "educated guess" if existing rules are insufficient for producing a rigorous answer.

We extend MetaQNL with soft matching—relaxing the rigid matching conditions when applying rules. Think of applying a rule $r$ to a set of assumptions $A$ as instantiating concrete rules: with rigid matching, we instantiate concrete rules $cr$ such that $cr$'s premises are $A$ and $cr$ must be an instance of $r$. In contrast, soft matching produces both concrete rules $cr$ and scores. $cr$'s premises are still $A$ but $cr$ is not required to be an instance of $r$. Further, the matching scores can be aggregated to calculate proof scores, allowing us to produce fuzzy proofs when rigorous proofs are impossible.

**Definition 5.1** (Soft matching). Given a rule $r$ and concrete sentences $A = \{a_1, a_2, \ldots, a_n\}$ as assumptions, soft matching outputs concrete rules with scores: $(cr_1, s_1), (cr_2, s_2), \ldots, (cr_k, s_k)$ such that (1) $\forall i, s_i \in [0, 1]$. (2) $\forall i, cr_i$'s premises are $A$.

There are many possible ways to realize soft matching, including using neural networks. In this paper, as an initial step, we perform soft matching only in testing. During training, MetaInduce still uses rigid matching for learning rules. Once the rules have been learned, we use them for making predictions with soft matching. We study two simple mechanisms for soft matching based on:

- *Neural pretrained language model*: Given $r$ and $A$, we use a pretrained language model to output concrete rules. Specifically, we encode soft matching as a seq2seq task and finetune a T5 (Raffel et al., 2020) model. Note that it requires supervision for finetuning.

- *Symbolic anti-unification*: Given $r$ and $A$, $r$ is not applicable, but we can find a more general applicable rule $r'$ by anti-unifying $A$ with $r$'s premises. For example, anti-unifying "`An elephants smiles`" with "`The [A] smiles`" produces "`[A] smiles`". Note that rigid matching is a special case when $r$ itself is applicable. For calculating the matching score, we use heuristics based on the number of perfectly matched words between $r$ and $A$.

## 6 Experiments

We instantiate MetaQNL/MetaInduce on three tasks: learning compositional instructions on MiniS-CAN (Lake et al., 2019)/SCAN (Lake & Baroni, 2018), logical reasoning on RuleTaker (Clark et al., 2020), and morphological analysis on SIGMORPHON 2018 (Cotterell et al., 2018). We achieve state-of-the-art accuracies on the three synthetic datasets (MiniSCAN, SCAN, and RuleTaker) using only a minor fraction of training data. On MiniSCAN and SCAN, MetaInduce recovers the ground truth rules precisely. Further, we evaluate our method with soft matching on two non-synthetic datasets: SIGMORPHON 2018 and ParaRules. Results show the promise of our method in handling noise and ambiguity that are ubiquitous in real-world data. Note that our experiments do not show MetaQNL/MetaInduce readily outperform neural

networks on *unconstrained, free-form natural language.* Nevertheless, they demonstrate the promise of a radically different learning approach.

### 6.1 Learning Compositional Instructions

**Task and Dataset.** MiniSCAN and SCAN have a similar format of translating a source sequence to a target sequence, e.g., "`jump` → `JUMP`", "`jump twice` → `JUMP JUMP`". MiniSCAN consists of only 14 training examples, whereas SCAN has 17K. State-of-the-art methods have reached 100% accuracy on both datasets (Liu et al., 2020; Nye et al., 2020; Chen et al., 2020).

**Experimental Setup.** In training, each source/target pair $x \to y$ is treated as a provable example $(A_i, g_i)$, with empty assumptions $A_i = \varnothing$ and the goal $g_i =$ "$x$ `$MAPS_TO$` $y$", for examples:

$$
\begin{array}{rcl}
\texttt{walk} & \texttt{\$MAPS\_TO\$} & \texttt{WALK} \\
\texttt{turn right} & \texttt{\$MAPS\_TO\$} & \texttt{RIGHT} \\
\texttt{jump after turn left} & \texttt{\$MAPS\_TO\$} & \texttt{LEFT JUMP} \\
\texttt{turn left twice} & \texttt{\$MAPS\_TO\$} & \texttt{LEFT LEFT} \\
\texttt{turn around right} & \texttt{\$MAPS\_TO\$} & \texttt{RIGHT RIGHT RIGHT RIGHT}
\end{array}
$$

In testing, we use "$x$ `$MAPS_TO$` `[Y]`" as the goal, where `[Y]` is a placeholder to be filled by the prover. The prover succeeds if it proves a goal with any `[Y]`. There is no unprovable example.

We use a rule proposer independent of specific training examples. First, it generates all concrete rules with $\leq 2$ premises by combining the sentences in the training set in all possible ways. Then it filters the rules using prior knowledge about compositional generalization: The meaning of a long sequence depends on its subsequences. For example. "`jump $MAPS_TO$ JUMP` ⊢ `jump twice $MAPS_TO$ JUMP JUMP`" is a valid rule, since `jump` is a subsequence of `jump twice`. But "`look $MAPS_TO$ LOOK` ⊢ `jump twice $MAPS_TO$ JUMP JUMP`" is not a valid rule. Similar assumptions were also made in the "interpretation grammar" in Nye et al. (2020) and the "analytical expressions" in Liu et al. (2020). Below are examples of concrete rules generated. Note that many of them do not make sense. However, we do not require the rule proposer to be accurate.

$$
\begin{array}{rcl}
\texttt{walk \$MAPS\_TO\$ WALK; run \$MAPS\_TO\$ RUN} & \vdash & \texttt{walk after run \$MAPS\_TO\$ RUN WALK} \\
\texttt{run \$MAPS\_TO\$ RUN} & \vdash & \texttt{walk after run \$MAPS\_TO\$ RUN WALK} \\
\texttt{run \$MAPS\_TO\$ RUN} & \vdash & \texttt{jump twice after run twice \$MAPS\_TO\$} \\
& & \qquad \texttt{RUN RUN JUMP JUMP} \\
\texttt{run twice \$MAPS\_TO\$ RUN RUN} & \vdash & \texttt{jump twice after run twice \$MAPS\_TO\$} \\
& & \qquad \texttt{RUN RUN JUMP JUMP}
\end{array}
$$

We use backward chaining as the prover and Z3 (De Moura & Bjørner, 2008) as the MAX-SAT solver. For SCAN, we train only on the 400 shortest examples and test on four different splits: `simple`, `length`, `addprim_jump`, and `addprim_turn_left`.

**Results.** On both datasets, MetaInduce achieves 100% testing accuracy and successfully recovers the ground truth rules. Below are example rules learned from SCAN (more in Appendix D):

```
                                            ⊢  walk $MAPS_TO$ WALK
                                            ⊢  turn right $MAPS_TO$ RIGHT
                                            ⊢  turn left $MAPS_TO$ LEFT
                                            ⊢  turn around left $MAPS_TO$
                                                  LEFT LEFT LEFT LEFT
                                            ⊢  turn around right $MAPS_TO$
                                                  RIGHT RIGHT RIGHT RIGHT
                  [A] $MAPS_TO$ [B]  ⊢  [A] left $MAPS_TO$ LEFT [B]
                  [A] $MAPS_TO$ [B]  ⊢  [A] right $MAPS_TO$ RIGHT [B]
                  [A] $MAPS_TO$ [B]  ⊢  [A] twice $MAPS_TO$ [B] [B]
                  [A] $MAPS_TO$ [B]  ⊢  [A] thrice $MAPS_TO$ [B] [B] [B]
   [A] $MAPS_TO$ [B]; [C] $MAPS_TO$ [D]  ⊢  [C] and [A] $MAPS_TO$ [D] [B]
   [A] $MAPS_TO$ [B]; [C] $MAPS_TO$ [D]  ⊢  [A] after [C] $MAPS_TO$ [D] [B]
```

Our experiments take 30 minutes to run on a laptop, which compares favorably with methods using deep neural networks (e.g., 1 day on GPUs in Liu et al. (2020)).

In Eqn. 1, smaller $\lambda^+$ encourages more compact models (i.e., a small number of learned rules) potentially at the cost of accuracy. We tune $\lambda^+$ on 1000 validation examples. Results in Table 1 show that the accuracy on SCAN is fairly robust w.r.t. $\lambda^+$.

Table 1: Validation accuracies on SCAN with different $\lambda^+$.

| $\lambda^+$ | 0.32 | 0.64 | 1.28 | 2.56 | 5.12 | $\infty$ |
|---|---|---|---|---|---|---|
| #Rules learned | 16 | 17 | 20 | 20 | 20 | 20 |
| Accuracy | 85.9 | 90.3 | 100.0 | 100.0 | 100.0 | 100.0 |

## 6.2 Logical Reasoning

**Task and Experimental Setup.** RuleTaker tests logical reasoning using synthetic sentences. It consists of examples similar to Fig. 1. We use the OWA (open-world assumption) version introduced by Tafjord et al. (2021), where a sentence can be proved, disproved, or neither. For example, if "`The elephant be tall`" is true, then "`The elephant be not tall`" should be false. To handle such cases, we prepend sentences with special symbols $TRUE$ or $FALSE$, so that the example can be disproved using the rule "`$TRUE$ The elephant be tall ⊢ $FALSE$ The elephant be not tall`". For each example to be proved, we add it to the set of provable examples $\mathcal{D}_{\text{train}}^+$ and its negation to unprovable examples $\mathcal{D}_{\text{train}}^-$. Conversely, for each example to be disproved, we add it to $\mathcal{D}_{\text{train}}^-$ and its negation to $\mathcal{D}_{\text{train}}^+$. For examples that can be neither proved nor disproved, we add both itself and its negation to $\mathcal{D}_{\text{train}}^-$.

RuleTaker includes ground truth proofs providing concrete rules such as "`$TRUE$ The elephant be tall ⊢ $FALSE$ The elephant be not tall`" but not any abstraction that allows generalizing beyond the specific examples. Our rule proposer simply generates these ground truth concrete rules, whereas MetaInduce tries to learn abstract rules. And we use simple heuristics for filtering invalid rules generated by anti-unification (Appendix F). Note that prior works on RuleTaker also rely on ground truth proofs (Clark et al., 2020; Saha et al., 2020; Tafjord et al., 2021).

On machines with 0 GPUs, 32GB RAM, and 4 CPUs, we run MetaInduce for 5 epochs on 10K training examples, which takes about 20 hours. We use forward chaining as the prover and a depth limit of 7. The

hyperparameters $\lambda^+$ and $\lambda^-$ are tuned on validation data. Examples of learned rules are in Appendix E. Our experiments take 2 days. It is impossible to directly compare our run time with ProofWriter since their code is not publicly available.

Table 2: Answer predicting accuracies on RuleTaker (OWA). Rows are depths of test proofs. N/A means the test example can be neither proved nor disproved.

|  | D3 | | D5 | |
| --- | --- | --- | --- | --- |
| | ProofWriter | Ours | ProofWriter | Ours |
| N/A | **99.9** | 99.4 | 99.4 | **99.6** |
| 0 | 100.0 | 100.0 | 100.0 | 100.0 |
| 1 | 99.3 | **100.0** | 100.0 | 100.0 |
| 2 | 99.7 | 99.7 | 99.9 | **100.0** |
| 3 | **99.2** | 98.9 | 100.0 | 100.0 |
| 4 | **99.1** | 98.9 | **99.9** | 99.4 |
| 5 | **98.8** | 98.6 | **100.0** | 99.1 |
| All | **99.6** | 99.4 | 99.7 | 99.7 |

Table 3: Answer predicting accuracies on the OWA version of ParaRules. The model is trained on D3 + ParaRules and tested on ParaRules.

|  | PRover | ProofWriter | Ours |
| --- | --- | --- | --- |
| 0 | 99.7 | 99.9 | **100.0** |
| 1 | 98.6 | 99.3 | **99.7** |
| 2 | 98.2 | 98.3 | **99.4** |
| 3 | 96.5 | 98.2 | **98.8** |
| 4 | 88.0 | 91.5 | **100.0** |
| All | 98.4 | 99.1 | **99.7** |

**Results on Synthetic Data.** The RuleTaker dataset consists of simple synthetic sentences generated by templates. We compare our method with ProofWriter (Tafjord et al., 2021)—a state-of-the-art method that also uses ground truth proofs. Following their setup, we test on D5 (a subset of RuleTaker with proof depths $\leq 5$) and train separate models on D5 and D3 (proof depths $\leq 3$). Results are in Table 2. MetaInduce achieves state-of-the-art accuracies and is competitive with ProofWriter. Further, it learns significantly more compact models with much less training data. For example, the model trained on D3 with $\lambda^+ = \lambda^- = 1.28$ using only 14% of the training data has only 79 rules and a total of 2869 symbols, but achieves an overall test accuracy of 99.4%. In comparison, ProofWriter has an accuracy of 99.6% and is based on T5-11B (Raffel et al., 2020), which has 11 billion parameters.

Note that we only compare with neural approaches, as many existing neurosymbolic approaches are not readily applicable to our task setting without significant manual design. For example, Rocktäschel & Riedel (2017); Evans & Grefenstette (2018); Cohen et al. (2018); Saparov & Mitchell (2022) require human-defined predicates. Lee et al. (2016); Weber et al. (2019) require semantic parsing.

**Handling Noisy Data with Soft Matching.** To test our system using data closer to natural language, we evaluate on ParaRules—a part of RuleTaker consisting of 40K examples paraphrased into natural language by crowd workers. We apply the language model–based soft matching mechanism described in Sec. 5. Following ProofWriter, we train on D3 + ParaRules and test on ParaRules.

First, we run MetaInduce on (synthetic) D3 to learn a set of rules just like before. Let $T$ be the set of sentences in the learned rules. Then we extract a dataset from ParaRules for training the soft matching network. ParaRules contains pairs of $(s, p)$, where $s$ is a synthetic sentence ("`The elephant like cats`") and $p$ is its paraphrased version (e.g., "`The elephant is fond of felines`"). For each $(s, p)$ and $t \in T$

Table 4: F1 scores of morphological analysis. *FP* is the average of FUT and PST. *O* represents OTHER.

| Model | Spanish | | Swahili | | Turkish | |
|---|---|---|---|---|---|---|
| | FP | O | FP | O | FP | O |
| LSTMs + Attention | 66 | **88** | 75 | **90** | **69** | **85** |
| Ours | 55 | 82 | **81** | 86 | 53 | 71 |
| Ours + Soft matching | 66 | 84 | 80 | 85 | 53 | 70 |

(e.g., "`[A] [B]`"), we check whether $s$ can match with $t$ symbolically (yes in this case). If so, we know applying soft matching to $p$ and $t$ should produce $s$. And the training data consists of all triples of $(s, p, t)$.

The soft matching network is implemented by finetuning a T5 model (Raffel et al., 2020). The input sequence concatenates $p$ and $t$ ("`the elephant is fond of felines <SEP> [A] [B]`"). And the output is $s$ ("`the elephant like cats`"). We finetune the model with a learning rate of $10^{-4}$ and a batch size of 32 using the AdamW optimizer (Loshchilov & Hutter, 2019). Once trained, we use the soft matching network to perform matching when evaluating the learned rules on ParaRules. Results in Table 3 show an overall accuracy of 99.7%, outperforming PRover (Saha et al., 2020) and ProofWriter.

### 6.3 Morphological Analysis

**Task and Dataset.** Finally, we evaluate on the morphological analysis task in Akyürek et al. (2021). Morphological analysis is not a reasoning task. But it is frequently used by prior work as a testbed of compositional generalization and learning symbolic rules/programs from data, which are two important aspects of our method. Given the surface form of a word (e.g., `studied`), the model predicts its lemma (`study`) and an unknown number of tags, such as `SG` (singular) and `PST` (past tense). The data is constructed from the SIGMORPHON 2018 dataset. It consists of 3 languages with varying morphological complexity—Spanish, Swahili, and Turkish. For each language, they sample a training set of 1K examples and three test sets of 100 examples each (FUT, PST, and OTHER). FUT consists exclusively of words in the future tense; PST consists of words in the past tense. The training set has only 8 past-tense words and 8 future-tense words. Therefore, FUT and PST test models' few-shot learning capabilities.

**Experimental Setup.** To apply MetaQNL, we represent both the surface form and the lemma as characters. The surface form serves as the assumption, whereas the lemma and the tags serve as conclusions. For example, for the Spanish surface form `zarandeamos` with lemma `zarandear` and tags `V;IND;PRS;1;PL`, we treat `z a r a n d e a m o s` as the assumptions and construct 6 provable examples with goals `$LEMMA$ z a r a n d e a r`, `$TAG$ V`, $\cdots$, `$TAG$ PL`. Any other goals are treated as unprovable. The rule proposer simply generates rules that can prove the conclusion in a single step. For this example, it generates the 6 candidate rules below:

```
z a r a n d e a m o s ⊢ $LEMMA$ z a r a n d e a r
z a r a n d e a m o s ⊢ $TAG$ V
z a r a n d e a m o s ⊢ $TAG$ IND
z a r a n d e a m o s ⊢ $TAG$ PRS
z a r a n d e a m o s ⊢ $TAG$ 1
z a r a n d e a m o s ⊢ $TAG$ PL
```

Following Akyürek et al. (2021), we evaluate the predictions using F1 score and compare with a standard seq2seq neural network: LSTMs (Hochreiter & Schmidhuber, 1997) with attention. Note that we're comparing with the baseline in Akyürek et al. (2021), not their proposed method. Their method is orthogonal to us since it focuses on augmenting the training data.

**Results w/o Soft Matching.** Results are shown in Table 4. Note that the task is not trivial: the neural baseline performs far from perfect, especially on FUT and PST (an F1 score of 66% on Spanish). Our method (without soft matching) is competitive with the baseline on Swahili, but there are still gaps on Spanish and Turkish. Further analysis reveals different reasons behind the gaps: Turkish morphology is very complex. But our current way of instantiating MetaQNL only considers proofs of depth 1, which could be a restriction for learning more expressive rules. Spanish morphology is relatively simple. Our F1 score still has a large room for improvement, because it learns over-specific rules that achieve high precision but low recall.

Below are example rules learned from the Spanish part of the dataset. Unlike the baseline, our method learns interpretable morphological rules; e.g., the suffix áramos indicates the past tense.

```
        [A] e a m o s  ⊢  $LEMMA$ [A] e a r
        [A] á r a m o s  ⊢  $TAG$ PST
    n o s - [A] e m o s  ⊢  $TAG$ SBJV
 n o - [A] z c a m o s  ⊢  $LEMMA$ [A] c e r
        [A] z a m o s  ⊢  $LEMMA$ [A] z a r
```

**Results w/ Soft Matching.** To explore soft matching, we keep training the same and apply soft matching only in testing. Given a testing example such as "z a r a n d e a m o s", we consider not only applicable rules learned by MetaInduce but also additional rules generated through our simple soft matching mechanism based on anti-unification (Sec. 5). All rules are ranked based on their matching scores, which are calculated using heuristics. Rigid matching always has the highest score, and more perfectly matched characters lead to higher scores. After ranking the rules, we apply them one by one until we get a predicted lemma.

The bottom row in Table 4 shows the result of soft matching. Even this simple form of soft matching can close the gap on Spanish. However, it leads to no improvements on Swahili and Turkish. We found that the individual rules learned on Swahili and Turkish are more approximate, i.e. more like "rules of thumb"—they capture the general pattern but have many exceptions. This is due to the increased morphological complexity. In these two languages, there are fewer simple rules such as "This suffix always indicates the past tense." As a result, relaxing the matching conditions naively would lead to too many spurious rules.

# 7 Limitations and Open Questions

First, our approach is far from mature. Substantial further development is needed for handling free-form natural language, e.g., in benchmarks of arithmetic or commonsense reasoning (Cobbe et al., 2021; Talmor et al., 2019). Soft matching is one possible way to address linguistic variations, e.g., by using a pretrained language model to output matching scores between rules and assumptions.

Our experiments are not large-scale but serve as proof of concept for a novel approach at an early stage. MetaInduce does not yet scale to millions of training examples (Sec. 4.3), which may be necessary to learn enough rules to handle the complexity of natural language. The current bottleneck is rule abstraction, which can be possibly addressed through better methods than anti-unification.

MetaInduce is a meta algorithm that permits many variations of its components. This provides many open questions and opportunities for integration with deep learning. For example, the rule proposer or theorem prover can be a deep network instead of a manually crafted heuristic.

**Acknowledgments**

This work is partially supported by the Office of Naval Research under Grant N00014-20-1-2634. The authors also gratefully acknowledge financial support from the Schmidt DataX Fund at Princeton University, made possible through a major gift from the Schmidt Futures Foundation.

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

# A   Partial Order Among Sentences and Rules

Here we prove that the $\leq$ in Definition 3.4 is indeed a partial order relation. First note that the equality between rules is defined modulo premise reordering and $\alpha$-conversion. In other words, premises are unordered, and variable renaming does not change a rule.

**Definition A.1** (Sentence length). The length of $s = (t_1, t_2, \ldots, t_n) \in \Sigma^+$, denoted by $\text{length}(s)$, is $n$.

**Lemma A.2** (Substitutions are noncontractive). *Applying substitutions does not make a sentence shorter. More formally, for any sentence $s = (t_1, t_2, \ldots, t_n) \in \Sigma^+$ and substitution $\sigma : \Sigma_v \to \Sigma_{-s}^+$, we have $length(\sigma s) \geq n$. Further, $length(\sigma s) = n$ if and only if $\sigma$ maps all tokens in $s$ to sentences of length 1, i.e., $\forall i, length(\sigma t_i) = 1$.*

*Proof.* For any substitution $\sigma : \Sigma_v \to \Sigma_{-s}^+$ and variable $v \in \Sigma_v$, $\sigma(v) \in \Sigma_{-s}^+$ is a sentence. Therefore, for any token $t \in \Sigma$, $\text{length}(\sigma t) \geq 1$ (Definition 3.3). For any sentence $s = (t_1, t_2, \ldots, t_n)$, $\text{length}(\sigma s) = \sum_{i=1}^{n} \text{length}(\sigma t_i) \geq n$. And the equality holds if and only if $\forall i, \text{length}(\sigma t_i) = 1$. $\qquad\square$

**Theorem A.3** (Partial order among sentences). *If sentence equality is defined modulo $\alpha$-conversion, then the $\leq$ in Definition 3.4 is a partial order among sentences. In other words,*

1. *$\forall s \in \Sigma^+, s \leq s$.*

2. *$\forall s_1, s_2 \in \Sigma^+$, if $s_1 \leq s_2$ and $s_2 \leq s_1$, then $s_1 = s_2$ modulo $\alpha$-conversion.*

3. *$\forall s_1, s_2, s_3 \in \Sigma^+$, if $s_1 \leq s_2$ and $s_2 \leq s_3$, then $s_1 \leq s_3$.*

*Proof.* We prove the 3 statements separately.

1. Let $\epsilon$ be the identity substitution mapping any variable to itself, i.e., $\forall v \in \Sigma_v, \epsilon(v) = v$. From Definition 3.3, $\epsilon$ also maps any token to itself ($\forall t \in \Sigma, \epsilon t = t$), and therefore any sentence to itself ($\forall s \in \Sigma^+, \epsilon s = s$). Applying Definition 3.4, we have $\forall s \in \Sigma^+, s \leq s$.

2. Given two sentences $s_1 = (t_1, t_2, \ldots, t_n)$, $s_2 = (t_1', t_2', \ldots, t_m')$ such that $s_1 \leq s_2$ and $s_2 \leq s_1$, there exist substitutions $\sigma$, $\varphi$ such that $s_1 = \sigma s_2$ and $s_2 = \varphi s_1$ (Definition 3.4). Applying Lemma A.2 to them separately leads to $n = m$ and $\forall i, \text{length}(\varphi t_i) = \text{length}(\sigma t_i') = 1$. According to Definition 3.3, we derive $\forall i, t_i = \sigma t_i'$ and $t_i' = \varphi t_i$. If $t_i$ is not a variable, $t_i' = \varphi t_i = t_i$, i.e., all non-variable tokens in $s_1$ and $s_2$ are identical. If $t_i$ is a variable, $t_i'$ must also be a variable because otherwise $t_i = \sigma t_i'$ would not be a variable. Therefore, both $\sigma$ and $\varphi$ are just renaming variables. And it is straightforward to verify that they cannot map different variables to the same. In other words, $\sigma$ and $\varphi$ are $\alpha$−conversions; $s_1 = s_2$ modulo $\alpha$-conversion.

3. Given three sentences $s_1$, $s_2$, and $s_3$ such that $s_1 \leq s_2$ and $s_2 \leq s_3$, there exist substitutions $\sigma$ and $\varphi$ such that $s_1 = \sigma s_2$ and $s_2 = \varphi s_3$. Let $\mu = \sigma \circ \varphi$ be the function composite of $\sigma$ and $\varphi$. $\mu$ is also a substitution and $s_1 = \mu s_3$. Therefore, $s_1 \leq s_3$.

$\qquad\square$

**Theorem A.4** (Partial order among rules). *The $\leq$ in Definition 3.4 is a partial order among rules. In other words,*

1. *For any rule $r$, $r \leq r$.*

2. *For any two rules $r_1$ and $r_2$, if $r_1 \leq r_2$ and $r_2 \leq r_1$, then $r_1 = r_2$ modulo $\alpha$-conversion.*

3. *For any three rules $r_1$, $r_2$ and $r_3$, if $r_1 \leq r_2$ and $r_2 \leq r_3$, then $r_1 \leq r_3$.*

*Proof.* Similar to the proof of Theorem A.3. $\qquad\square$

**Definition A.5** (Strictly partial order among sentences/rules). Let $s_1$ and $s_2$ be two sentences, $s_1$ is strictly more general than $s_2$ (denoted by $s_2 < s_1$) if and only if $s_2 \le s_1$ and $s_1 \ne s_2$ modulo $\alpha$-conversion. Similarly, if $r_1$ and $r_2$ are rules, $r_2 < r_1$ if and only if $r_2 \le r_1$ and $r_1 \ne r_2$ modulo $\alpha$-conversion.

# B    Turing Completeness of MetaQNL

We prove the Turing completeness of MetaQNL (Theorem 3.7) through its connection with unrestricted grammars, which have been known to be equivalent to Turing machines. First, we repeat a few definitions that can be found in common textbooks on formal languages and automata theory (Hopcroft et al., 2001).

**Definition B.1** (Unrestricted grammar). An unrestricted grammar is a formal grammar $G = (N, T, P, S)$, where $N$ is a finite set of nonterminal symbols, $T$ is a finite set of terminal symbols, and $N \cap T = \varnothing$. $P$ is a finite set of production rules of the form $\alpha \to \beta$, where $\alpha \in (N \cup T)^+$ is an non-empty string, and $\beta \in (N \cup T)^*$ is an arbitrary string. $S \in N$ is a designated start symbol.

**Definition B.2** (Direct derivation). For any strings $u, v \in (N \cup T)^*$, we say $u$ directly derives to $v$ (denoted as $u \Rightarrow v$) if and only if there exists a production rule $\alpha \to \beta$ such that $u = l\alpha r$ and $v = l\beta r$, where $l, r \in (N \cup T)^*$ are arbitrary strings.

**Definition B.3** (Derivation). The "derives to" relation ($\Rightarrow^*$) is the reflexive transitive closure of the "direct derives to" relation ($\Rightarrow$).

**Definition B.4** (Language of unrestricted grammar). Given an unrestricted grammar $G = (N, T, P, S)$, its language $L(G) = \{w \in T^* \mid S \Rightarrow^* w\}$ consists of all terminal strings derivable from the start symbol $S$.

Next, we prove that for any unrestricted grammar $G$, there exists a set of MetaQNL rules that can recognize its language $L(G)$.

**Theorem B.5** (MetaQNL can express unrestricted grammar). *For any unrestricted grammar $G = (N, T, P, S)$, there exists a set of MetaQNL rules that can recognize $L(G)$, i.e., for any $w \in T^*$, $w \in L(G)$ if and only if $S \Rightarrow^* w$ is provable in MetaQNL.*

*Proof.* Let $G = (N, T, P, S)$ be an arbitrary unrestricted grammar. We construct a system of MetaQNL rules with the word vocabulary $\Sigma_w = N \cup T$, the variable vocabulary $\Sigma_v = \{\text{L}, \text{R}\}$, and the special symbol vocabulary $\Sigma_s = \{\Rightarrow^*\}$. We can assume they are disjoint without loss of generality. Then for each production rule $\alpha \to \beta \in P$, we construct four MetaQNL rules:

$$S \Rightarrow^* \text{[L]}\alpha\text{[R]} \quad \vdash \quad S \Rightarrow^* \text{[L]}\beta\text{[R]} \tag{2}$$

$$S \Rightarrow^* \alpha\text{[R]} \quad \vdash \quad S \Rightarrow^* \beta\text{[R]} \tag{3}$$

$$S \Rightarrow^* \text{[L]}\alpha \quad \vdash \quad S \Rightarrow^* \text{[L]}\beta \tag{4}$$

$$S \Rightarrow^* \alpha \quad \vdash \quad S \Rightarrow^* \beta. \tag{5}$$

And we have a special rule without any assumption:

$$\vdash S \Rightarrow^* S. \tag{6}$$

We want to prove that these rules can be used to recognize $L(G)$.

Let $w \in L(G)$, i.e., $S \Rightarrow^* w$. From Definition B.3, we have $S \Rightarrow u_1 \Rightarrow u_2 \Rightarrow \cdots \Rightarrow u_{k-1} \Rightarrow w$, where $u_i \in (N \cup T)^*$ are intermediate strings for deriving $w$. We prove that $S \Rightarrow^* w$ is provable in MetaQNL by induction on $k$. When $k = 1$, we have $S \Rightarrow w$. According to Definition B.2, there must be a production rule $S \to w \in P$. It corresponds to a MetaQNL rule $S \Rightarrow^* S \vdash S \Rightarrow^* w$ (Eqn. 5). Together with the special rule $\vdash S \Rightarrow^* S$ (Eqn. 6), we can prove $S \Rightarrow^* w$ in MetaQNL.

When $k > 1$, since $u_{k-1} \Rightarrow w$, there must be a production rule $\alpha \to \beta \in P$ such that $u_{k-1} = l\alpha r$ and $w = l\beta r$, where $l, r \in (N \cup T)^*$. Regardless of whether $l$ and $r$ are empty. One of the four corresponding MetaQNL rules can be instantiated to be $S \Rightarrow^* l\alpha r \vdash S \Rightarrow^* l\beta r$, which is equivalent to $S \Rightarrow^* u_{k-1} \vdash S \Rightarrow^* w$. Also, the inductive hypothesis tells us $S \Rightarrow^* u_{k-1}$ is provable in MetaQNL. Therefore, $S \Rightarrow^* w$ is also provable.

Conversely, given any $w \in (N \cup T)^*$ such that $S \Rightarrow^* w$ is provable in MetaQNL, we want to prove $S \Rightarrow^* w$. We perform induction on the number of steps $k$ it takes to prove $S \Rightarrow^* w$ in MetaQNL. When $k = 1$, it must be proved by a single application of the special rule $\vdash S \Rightarrow^* S$ (Eqn. 6). Therefore $w = S$. According to Definition B.3, $S \Rightarrow^* w$ holds.

When $k > 1$, our proof depends on which one of the four types of MetaQNL rules (Eqn. 2 to Eqn. 5) is applied in the last step when proving $S \Rightarrow^* w$. Here we only consider Eqn. 2 since it is the most sophisticated case; other cases are similar. So $S \Rightarrow^*$ `[L]`$\alpha$`[R]` $\vdash S \Rightarrow^*$ `[L]`$\beta$`[R]` is the last MetaQNL rule applied, and $l, r \in (N \cup T)^+$ are the strings used to instantiate the variables `[L]` and `[R]` ($w = l\beta r$). By that time, we must have already proved the assumption $S \Rightarrow^* l\alpha r$ within $k - 1$ steps. So the inductive hypothesis tells us $S \Rightarrow^* l\alpha r$. The last MetaQNL rule corresponds to a production rule $\alpha \to \beta$. Definition B.2 tells us $l\alpha r \Rightarrow l\beta r$. Therefore, we have $S \Rightarrow^* l\beta r$, and $S \Rightarrow^* w$. $\qquad\square$

Theorem 3.7 is a direct corollary of Theorem B.5 since unrestricted grammars can express any recursively enumerable language.

## C   Unification and Anti-unification of Sentences and Rules

Unification and anti-unification (Plotkin, 1970; Robinson & Voronkov, 2001) are basic symbolic procedures in formal logic that are useful for theorem proving and logic programming (Russell & Norvig, 2002; Yernaux & Vanhoof, 2019). In MetaQNL, unification is used in backward chaining, and anti-unification is used to abstract concrete rules into rules with variables. We adapt existing problem setups and algorithms from formal logic to MetaQNL. The algorithms we use for MetaQNL do not have theoretical guarantees as in formal logic, but they work well in practice. The anti-unifiers they compute may not satisfy the conditions of most specific anti-unifiers (Definition C.7). But strict anti-unification is not necessary for rule abstraction to work. In principle, all we need is a procedure for generating abstract rules from concrete ones.

**Unification.**   Given two sentences (or two rules), unification aims to find substitutions mapping them to the same sentence (or rule). Such substitutions are called unifiers. We extend unification to MetaQNL by adapting prior work, especially the unification algorithm developed by Kutsia (2002) for a variant of first-order logic with sequence variables and flexible arity symbols.

**Definition C.1** (Unifier)**.** A substitution $\sigma : \Sigma_v \to \Sigma^+_{-s}$ is a unifier of two sentences $s_1, s_2 \in \Sigma^+$ if and only if $\sigma s_1 = \sigma s_2$. Similarly, it is a unifier of two rules $r_1$ and $r_2$ if and only if $\sigma r_1 = \sigma r_2$.

Two sentences may have multiple unifiers. Taking $s_1 =$ `[X] is [Y]`, $s_2 =$ `The elphant [Z]` as an example, their unifiers include $\sigma = \{$`[X]` $\to$ `The elephant`, `[Z]` $\to$ `is [Y]`$\}$, $\varphi = \{$`[X]` $\to$ `The elephant`, `[Y]` $\to$ `tall`, `[Z]` $\to$ `is tall`$\}$, etc. Both $\sigma$ and $\varphi$ are valid unifiers, but they lead to different sentences when applied: $\sigma s_1 =$ `The elephant is [Y]`, $\varphi s_1 =$ `The elephant is tall`. We prefer $\sigma$ to $\varphi$ because it is more general; it does not introduce any new information not in $s_1$ and $s_2$. In contrast, we cannot infer the "`tall`" in $\varphi$. This is the intuition behind the concept of "most general unifiers".

**Definition C.2** (Most general unifier)**.** Let the substitution $\sigma$ be a unifier of sentencens $s_1$ and $s_2$, it is a most general unifier if and only if there is no unifier $\varphi$ of $s_1$ and $s_2$ such that $\sigma s_1 < \varphi s_1$.

In unification, we want to compute a set of most general unifiers, and we want the set to be minimal and complete. Below we define these concepts for sentences.

**Definition C.3** (Complete set of unifiers)**.** Let $\mathcal{U}$ be a set of unifiers of sentences $s_1$ and $s_2$, $\mathcal{U}$ is complete if and only if for any unifier $\varphi$ of $s_1$ and $s_2$, there exists a unifier $\sigma \in \mathcal{U}$, such that $\varphi s_1 \leq \sigma s_1$.

**Definition C.4** (Minimal set of unifiers)**.** Let $\mathcal{U}$ be a set of unifiers of sentences $s_1$ and $s_2$, $\mathcal{U}$ is minimal if and only if for any $\sigma, \varphi \in \mathcal{U}$, $\varphi s_1 \leq \sigma s_1$ implies $\sigma = \varphi$ (modulo $\alpha$-conversion).

**Definition C.5** (Minimal complete set of unifiers)**.** Let $\mathcal{U}$ be a set of unifiers of sentences $s_1$ and $s_2$, $\mathcal{U}$ is a minimal complete set of unifiers if and only if it is both minimal and complete.

The definitions for rules are parallel. Given two sentences (or two rules), the unification problem is to compute a minimal complete set of unifiers. The result can be empty (e.g., unifying "`hello world`" and

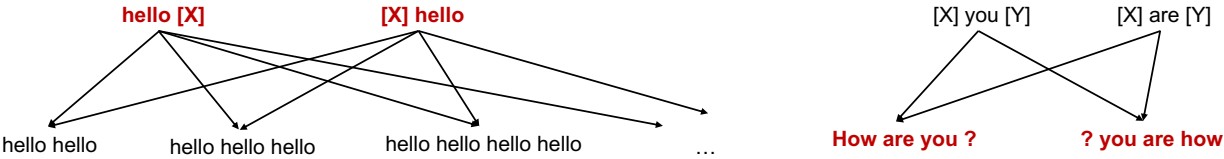

Figure 3: The minimal complete set of unifiers of two sentences can be empty, finite, or infinite (e.g., "`hello [X]`" and "`[X] hello`"). The minimal complete set of anti-unifiers is non-empty and finite.

"`how are you`"), finite ("`hello [X]`" and "`[Y] world`"), or infinite ("`hello [X]`" and "`[X] hello`", Fig. 3 *Left*).

**Anti-unification.** Given two sentences (or two rules), anti-unification aims to generalize them into a single sentence (or rule). Anti-unification has also been studied in formal logic (Plotkin, 1970; Kutsia et al., 2014). We extend it to MetaQNL by adapting prior work. For simplicity, we define anti-unification only for sentences, but it applies to rules as well.

**Definition C.6** (Anti-unifier). Given two sentences $s_1$ and $s_2$, their anti-unifier is a triple $(s, \sigma_1, \sigma_2)$ of a sentence $s$ and two subsitutions $\sigma_1$, $\sigma_2$, such that $\sigma_1 s = s_1$ and $\sigma_2 s = s_2$.

**Definition C.7** (Most specific anti-unifier). Let $(s, \sigma_1, \sigma_2)$ be an anti-unifier of sentencens $s_1$ and $s_2$, it is a most specific anti-unifier if and only if there is no substitution $\varphi$, $\sigma_1'$ and $\sigma_2'$ such that

1. $\sigma_1 = \sigma_1' \circ \varphi$, $\sigma_2 = \sigma_2' \circ \varphi$

2. $\varphi s < s$

3. $(\varphi s, \sigma_1', \sigma_2')$ is also an anti-unifier of $s_1$ and $s_2$

**Definition C.8** (Complete set of anti-unifiers). Let $\mathcal{A}$ be a set of anti-unifiers of sentences $s_1$ and $s_2$, $\mathcal{A}$ is complete if and only if for any anti-unifier $(s, \sigma_1, \sigma_2)$ of $s_1$ and $s_2$, there exists a substitution $\varphi$ and an anti-unifier $(\varphi s, \sigma_1', \sigma_2')$ such that $\sigma_1 = \sigma_1' \circ \varphi$, $\sigma_2 = \sigma_2' \circ \varphi$.

**Definition C.9** (Minimal set of anti-unifiers). Let $\mathcal{A}$ be a set of anti-unifiers of sentences $s_1$ and $s_2$, $\mathcal{A}$ is minimal if and only if for any $(s, \sigma_1, \sigma_2), (s', \sigma_1', \sigma_2') \in \mathcal{A}$, if there exists a substitution $\varphi$ such that

1. $s' = \varphi s$

2. $\sigma_1 = \sigma_1' \circ \varphi$, $\sigma_2 = \sigma_2' \circ \varphi$

then $\varphi$ must be an $\alpha$-conversion, i.e. $\varphi s = s$ (modulo $\alpha$-conversion).

**Definition C.10** (Minimal complete set of anti-unifiers). Let $\mathcal{A}$ be a set of anti-unifiers of sentences $s_1$ and $s_2$, $\mathcal{A}$ is a minimal complete set of anti-unifiers if and only if it is both minimal and complete.

Given two sentences (or two rules), the anti-unification problem is to compute a minimal complete set of anti-unifiers. Unlike unification, the result of anti-unification must be non-empty and finite (Fig. 3).

**Theorem C.11** (Anti-unification is finitary). *Let $\mathcal{A}$ be a minimal complete set of anti-unifiers of sentences $s_1$ and $s_2$, then $\mathcal{A}$ is non-empty and finite.*

*Proof.* For any sentences $s_1$ and $s_2$, we have a trivial anti-unifier $([X], \sigma_1, \sigma_2)$ where $\sigma_1 = \{[X] \to s_1\}$ and $\sigma_2 = \{[X] \to s_2\}$. Since $\mathcal{A}$ is complete, apply Definition C.8 and we will know $\mathcal{A}$ must be non-empty.

For any anti-unifier $(s, \varphi_1, \varphi_2) \in \mathcal{A}$, we have $\varphi_1 s = s_1$ (Definition C.6). Apply Lemma A.2 to derive $\text{length}(s) \leq \text{length}(\varphi_1 s) = \text{length}(s_1)$. Therefore, the length of $s$ is bounded. Also, $s$ cannot have non-variable tokens besides those in $s_1$, so its vocabulary is also bounded. There are a finite number of different sentences that $s$ can take. Therefore, $\mathcal{A}$ must also be finite. $\qquad\square$

Our current anti-unification algorithm is adapted from Kutsia et al. (2014). It recursively matches the beginning of two sentences. Let $s_1$ and $s_2$ be sentences, and $s$ is a more general sentence in their anti-unifier. If $s_1$ and $s_2$ start with the same word $w$, $s$ should also start with $w$. Otherwise, $s$ should start with a variable corresponding to some prefixes of $s_1$ and $s_2$. The algorithm searches for all such prefixes and anti-unifies the remaining parts of the sentences recursively.

## D   Details of MiniSCAN/SCAN Experiments

**MiniSCAN.**   The 14 MiniSCAN (Lake et al., 2019) training examples represented as sentences in MetaQNL (`$MAPS_TO$` is a special symbol):

```
                       dax  $MAPS_TO$  RED
                       lug  $MAPS_TO$  BLUE
                       wif  $MAPS_TO$  GREEN
                       zup  $MAPS_TO$  YELLOW
                   dax fep  $MAPS_TO$  RED RED RED
                   lug fep  $MAPS_TO$  BLUE BLUE BLUE
            wif blicket dax  $MAPS_TO$  GREEN RED GREEN
            lug blicket wif  $MAPS_TO$  BLUE GREEN BLUE
               dax kiki lug  $MAPS_TO$  BLUE RED
               lug kiki wif  $MAPS_TO$  GREEN BLUE
           lug fep kiki wif  $MAPS_TO$  GREEN BLUE BLUE BLUE
           lug kiki wif fep  $MAPS_TO$  GREEN GREEN GREEN BLUE
   wif kiki dax blicket lug  $MAPS_TO$  RED BLUE RED GREEN
   wif blicket dax kiki lug  $MAPS_TO$  BLUE GREEN RED GREEN
```

The 10 testing examples:

```
              zup fep  $MAPS_TO$  YELLOW YELLOW YELLOW
      zup blicket lug  $MAPS_TO$  YELLOW BLUE YELLOW
         zup kiki dax  $MAPS_TO$  RED YELLOW
     zup fep kiki lug  $MAPS_TO$  BLUE YELLOW YELLOW YELLOW
     wif kiki zup fep  $MAPS_TO$  YELLOW YELLOW YELLOW GREEN
 lug kiki wif blicket zup  $MAPS_TO$  GREEN YELLOW GREEN BLUE
zup blicket wif kiki dax fep  $MAPS_TO$  RED RED RED YELLOW GREEN YELLOW
zup blicket zup kiki zup fep  $MAPS_TO$  YELLOW YELLOW YELLOW YELLOW YELLOW YELLOW
      dax blicket zup  $MAPS_TO$  RED YELLOW RED
         wif kiki zup  $MAPS_TO$  YELLOW GREEN
```

Below are some examples of candidate rules generated by the rule proposer. Note that many of them are wrong because the premises are not sufficient to deduce the conclusion.

```
                                     ⊢  lug fep kiki wif $MAPS_TO$
                                            GREEN BLUE BLUE BLUE
                  dax $MAPS_TO$ RED  ⊢  wif blicket dax $MAPS_TO$ GREEN RED GREEN
                 lug $MAPS_TO$ BLUE  ⊢  lug kiki wif $MAPS_TO$ GREEN BLUE
                 lug $MAPS_TO$ BLUE  ⊢  lug fep $MAPS_TO$ BLUE BLUE BLUE
   dax $MAPS_TO$ RED; lug $MAPS_TO$ BLUE  ⊢  wif blicket dax kiki lug $MAPS_TO$
                                            BLUE GREEN RED GREEN
```

MetaInduce learns 7 rules corresponding to the ground truth rules of MiniSCAN:

```
                                            ⊢  dax $MAPS_TO$ RED
                                            ⊢  lug $MAPS_TO$ BLUE
                                            ⊢  wif $MAPS_TO$ GREEN
                                            ⊢  zup $MAPS_TO$ YELLOW
                         [A] $MAPS_TO$ [B]  ⊢  [A] fep $MAPS_TO$ [B] [B] [B]
    [A] $MAPS_TO$ [B]; [C] $MAPS_TO$ [D]  ⊢  [A] kiki [C] $MAPS_TO$ [D] [B]
    [A] $MAPS_TO$ [B]; [C] $MAPS_TO$ [D]  ⊢  [A] blicket [C] $MAPS_TO$ [B] [D] [B]
```

**SCAN.**   Some examples in SCAN (Lake & Baroni, 2018):

```
                    walk  $MAPS_TO$  WALK
                    jump  $MAPS_TO$  JUMP
              turn right  $MAPS_TO$  RIGHT
     jump after turn left  $MAPS_TO$  LEFT JUMP
              walk right  $MAPS_TO$  RIGHT WALK
           walk after run  $MAPS_TO$  RUN WALK
           turn left twice  $MAPS_TO$  LEFT LEFT
         turn opposite left  $MAPS_TO$  LEFT LEFT
          turn around right  $MAPS_TO$  RIGHT RIGHT RIGHT RIGHT
         walk around left  $MAPS_TO$  LEFT WALK LEFT WALK LEFT WALK LEFT WALK
```

Below are some examples of the candidate rules generated by the rule proposer.

```
                        run $MAPS_TO$ RUN  ⊢  walk after run $MAPS_TO$ RUN WALK
   walk $MAPS_TO$ WALK; run $MAPS_TO$ RUN  ⊢  walk after run $MAPS_TO$ RUN WALK
                        run $MAPS_TO$ RUN  ⊢  jump twice after run twice $MAPS_TO$
                                                RUN RUN JUMP JUMP
           run twice $MAPS_TO$ RUN RUN  ⊢  jump twice after run twice $MAPS_TO$
                                                RUN RUN JUMP JUMP
```

MetaInduce learns 20 rules corresponding to the ground truth rules of SCAN:

```
                                            ⊢  walk $MAPS_TO$ WALK
                                            ⊢  look $MAPS_TO$ LOOK
                                            ⊢  run $MAPS_TO$ RUN
                                            ⊢  jump $MAPS_TO$ JUMP
                                            ⊢  turn right $MAPS_TO$ RIGHT
                                            ⊢  turn left $MAPS_TO$ LEFT
                                            ⊢  turn opposite left $MAPS_TO$
                                                   LEFT LEFT
                                            ⊢  turn opposite right $MAPS_TO$
                                                   RIGHT RIGHT
                                            ⊢  turn around left $MAPS_TO$
                                                   LEFT LEFT LEFT LEFT
                                            ⊢  turn around right $MAPS_TO$
                                                   RIGHT RIGHT RIGHT RIGHT
                         [A] $MAPS_TO$ [B]  ⊢  [A] left $MAPS_TO$ LEFT [B]
                         [A] $MAPS_TO$ [B]  ⊢  [A] right $MAPS_TO$ RIGHT [B]
                         [A] $MAPS_TO$ [B]  ⊢  [A] opposite left $MAPS_TO$
                                                   LEFT LEFT [B]
                         [A] $MAPS_TO$ [B]  ⊢  [A] opposite right $MAPS_TO$
                                                   RIGHT RIGHT [B]
                         [A] $MAPS_TO$ [B]  ⊢  [A] around left $MAPS_TO$
                                                   LEFT [B] LEFT [B] LEFT [B] LEFT [B]
                         [A] $MAPS_TO$ [B]  ⊢  [A] around right $MAPS_TO$
                                                   RIGHT [B] RIGHT [B] RIGHT [B] RIGHT [B]
                         [A] $MAPS_TO$ [B]  ⊢  [A] twice $MAPS_TO$ [B] [B]
                         [A] $MAPS_TO$ [B]  ⊢  [A] thrice $MAPS_TO$ [B] [B] [B]
  [A] $MAPS_TO$ [B]; [C] $MAPS_TO$ [D]  ⊢  [C] and [A] $MAPS_TO$ [D] [B]
  [A] $MAPS_TO$ [B]; [C] $MAPS_TO$ [D]  ⊢  [A] after [C] $MAPS_TO$ [D] [B]
```

## E  Details of RuleTaker Experiments

Examples in RuleTaker are lemmatized and converted to lowercase. We also remove periods and insert a space before each comma. Fig. 4 shows the form of ground truth proofs in RuleTaker. For this specific example, our rule proposer would generate 2 candidate rules below:

```
                    $TRUE$ the elephant be big;
                    $TRUE$ the elephant be tall;
                    $TRUE$ big , tall things be strong;
                ⊢  $TRUE$ the elephant be strong
```

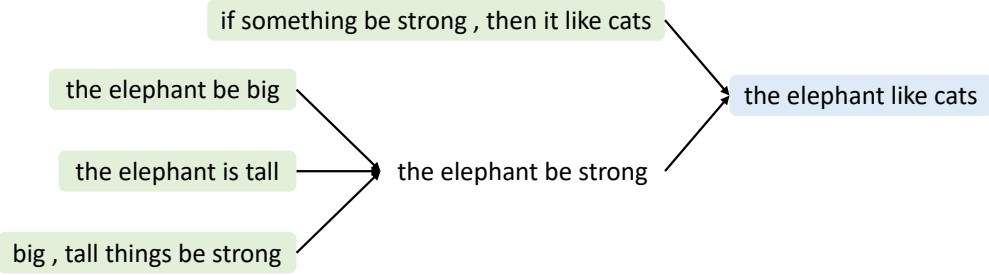

Figure 4: RuleTaker contains ground truth proofs in the form of directed acyclic graphs from the assumptions to the conclusion. The nodes in the graph are concrete sentences without variables.

```
        $TRUE$ the elephant be strong;
        $TRUE$ if something be strong , then it like cats;
  ⊢     $TRUE$ the elephant like cats
```

Below are some example rules learned from RuleTaker:

```
        $TRUE$ [A] [B];
        $TRUE$ [A] [C];
        $TRUE$ if someone [C] and they [B] then [D];
  ⊢     $TRUE$ [D]
```

```
            $TRUE$ [A] does not [B];
      ⊢     $FALSE$ [A] [B]
```

```
      $TRUE$ [A] [B];
      $TRUE$ [C] [D];
      $TRUE$ if someone [B] and [C] [D] then they [E];
  ⊢   $TRUE$ [A] [E]
```

```
            $TRUE$ [A] be [B];
            $TRUE$ [A] be [C];
            $TRUE$ [B] , [C] things be [E];
      ⊢     $TRUE$ [A] be [E]
```

```
      $TRUE$ [A] be [B];
      $TRUE$ [A] be [C];
      $TRUE$ if someone be [C] and [B] then they [D];
  ⊢   $TRUE$ [A] [D]
```

```
                    $TRUE$ [A] be [B] ⊢ $FALSE$ [A] be not [B]

                       $TRUE$ [A] [B];
                       $TRUE$ if something [B] then it [C]
                    ⊢  $TRUE$ [A] [C]
```

## F   Heuristics for constraining the space of rules

We use a few simple and general heuristics for constraining the space of rules and pruning invalid rules generated by anti-unification. First, we merge multiple variables that always appear together. For example, the [A] [B] [C] and [D] [E] in the rule below can be merged.

```
                       $TRUE$ if [A] [B] [C] then [D] [E];
                       $TRUE$ [A] [B] [C]
                    ⊢  $TRUE$ [D] [E]
```

So the rule becomes:

```
                       $TRUE$ if [A] then [B];
                       $TRUE$ [A]
                    ⊢  $TRUE$ [B]
```

A variable in a rule is called a *free variable*, if it appears only once. For example, the rule

```
             if something is red, then tomorrow will be sunny;
             [X] is red
          ⊢  tomorrow will be sunny
```

contains a free variable [X]. We only consider rules with no more than 1 free variable and require that they cannot appear in the conclusion. Because they would allow arbitrary conclusions formed by substituting them with other sentences. For example, the rule below is not allowed because of the free variable [X] in the conclusion:

```
             $TRUE$ today is sunny ⊢ $TRUE$ Tommorow is [X]
```

In addition, a rule cannot contain a premise made of one single free variable. Because this premise can be satisfied by any sentence, and there is no point including it in the rule. For example, the rule below is not allowed because of the free variable [X]:

```
                       $TRUE$ [X];
                       $TRUE$ if [A] then [B];
                       $TRUE$ [A]
                    ⊢  $TRUE$ [B]
```

