# OpenReview forum: "Learning Symbolic Rules for Reasoning in Quasi-Natural Language"
_TMLR — Accepted by TMLR_

### Review · Reviewer_5gc4 · 2023-01-29

**Summary Of Contributions:**

This paper presents a method for inducing a set of symbolic rules from examples of theorem proving in natural language.  The procedure, called MetaInduce, first involves proposing rules for individual examples, then "anti-unifying" those rules to abstract them, then finally pruning them using a MAX-SAT procedure. This procedure enforces that the rules should allow the data to be proven (softly; not every example must be provable) with a parsimonious set of rules.  At test time, a soft matching procedure is employed to be able to handle lexical divergences between assumptions and rules.

Results are given on three datasets: SCAN and MiniSCAN, RuleTaker, and a morphological analysis task from past work.  In the first two settings, the proposed method fits the data nearly perfectly, including on a paraphrased version of RuleTaker, and outperforms PRover and ProofWriter. On the third setting, the method shows mixed performance compared to an LSTM with attention.

As TMLR is about identifying claims and whether or not they are supported, I will try to identify the paper's claims here.  I believe them to be as follows:

Claim 1: "We propose MetaQNL, a symbolic system we call a "Quasi-Natural Language", which is compatible with not only rigorous logical inference but also natural language (Fig. 1)."

Claim 2: MetaQNL is Turing-Complete (proven in Appendix B; I won't discuss this claim further)

Claim 3: MetaInduce is a learning algorithm that can be solved efficiently by MAX-SAT solvers

Then various claims about the results, which I believe are all well-calibrated.

**Audience:**

Yes

**Claims And Evidence:**

Yes

**Requested Changes:**

Beyond possible modifications to the related work, I do not have any suggested changes for the paper.

**Strengths And Weaknesses:**

STRENGTHS

This paper is covering very exciting territory. The idea of inducing rules to enable logical reasoning in natural language is a very nice one. There is significant research in this space recently and I believe that this entry will be of interest to many researchers working here.

While aspects of the approach resemble those from prior work, I believe the rule selection using MAX-SAT is novel, as are the particulars of the rule abstraction piece. While it would've been nice to see some ablations to understand the impact of each of these components, the whole procedure together is novel and seems to work well.

It is nice to see results across three datasets. The generality of the approach is well-supported and the paper analyzes the results on each of the settings convincingly, with each setting bringing some new and different insight.

Based on the results, the claims this paper makes are well-supported.

WEAKNESSES

I believe the strengths in general outweigh the weaknesses for this paper. However, there are a few weaknesses.

The biggest weakness of this paper is that the main results are on artificial datasets. A procedure like MetaInduce/MetaQNL is likely to work well compared to neural networks when the underlying data reflects a regular inference process that MetaQNL can represent.  However, as the paper discusses in limitations, many approaches which work for RuleTaker do not generalize to other settings like EntailmentBank or other types of commonsense reasoning.

I do buy the argument that this is a first step and that it will take time to flesh out approaches like these and get them to the same level as more neural-first approaches. However, some of the challenges like scalability mentioned in 4.3 and the need for soft matching in 5 can quickly become unmanageable problems when carried to more realistic settings, and I question how easy it will be to address those shortcomings.  The neural theorem proving line of work has been around for a while, yet most of the winning neurosymbolic approaches look like slightly structured versions of neural inference (e.g., chain-of-thought) rather than NTPs. I don't see anything in this paper to really convince me that that state of affairs is going to change.

As a result, I think the low rule counts (a selling point of the results) don't reflect any magic properties of this approach, merely that these datasets can be fit with low description-length models. This is reflected by the fact that the model seems to struggle on Swahili and Turkish morphology and the soft matching doesn't alleviate this.

The related work section is generally quite complete. There are, however, a few recent papers particularly relevant to the idea of rule learning in natural language:

* Weir and van Durme Dynamic Generation of Interpretable Inference Rules https://arxiv.org/pdf/2209.07662.pdf
* Dalvi et al. Towards Teachable Reasoning Systems EMNLP 2022 https://arxiv.org/pdf/2204.13074.pdf
* Bostrom et al. Natural Language Deduction with Incomplete Information EMNLP 2022 https://arxiv.org/pdf/2211.00614.pdf
* Sanyal et al. Faithful and Robust Deductive Reasoning over Natural Language ACL 2022 https://arxiv.org/pdf/2203.10261.pdf

These are other recent papers in natural language reasoning:

* Creswell et al. Selection-Inference ICLR 2023 https://arxiv.org/abs/2205.09712
* Creswell and Shanahan Faithful Reasoning https://arxiv.org/abs/2205.09712

I do not think all of these necessarily need to cited, but I wanted to bring them to the authors' attention.

Furthermore, I think that this paper probably could go farther in comparing to other approaches. The systems above present many other ways of doing a similar kind of process, but this paper basically only compares to neural approaches.  When we already have a number of symbolic or neurosymbolic approaches for these tasks, why should a researcher prefer this framework? A better justification here would strengthen the results of the work.

---

> ### Author Response · Authors · 2023-02-18
> **Individual Response to Reviewer 5gc4**
>
> Thank you for your valuable feedback! Below we address your questions and concerns.
>
> ## Empirical Comparisons with Existing Neurosymbolic Approaches
>
> Please see the common response at the top.
>
> ## More Complex Tasks and Datasets
>
> Despite promising results, our method needs substantial future development for handling free-form natural language, e.g., in EntailmentBank or commonsense reasoning benchmarks, which contain significantly more linguistic variations. However, the main challenge might not be whether MetaQNL can represent the underlying inference process (given theoretical results on the expressiveness of MetaQNL, Theorem 3.7) but : (1) How to scale the method to learn more rules from more training examples. (2) A stronger soft matching mechanism for incorporating neural networks.
>
>
> ## Scalability
>
> While it is uncertain to what extent our method can be scaled up in the future, Sec. 4.3 includes a discussion of potential avenues.
>
>
> ## Soft Matching; Limitations on Purely Symbolic Approaches
>
> Instead of a limitation, we consider soft matching as an avenue for extending MetaQNLto more realistic tasks/datasets. The basic form of MetaQNL/MetaInduce (w/o soft matching) is a purely symbolic learning method. It demonstrates the merits of a learning paradigm drastically different from deep neural networks. However, we believe the future of machine learning might be neurosymbolic ([Garcez et al., 2020], [Gary, 2020]) instead of purely symbolic, as symbolic reasoning and neural networks have complementary strengths.
>
> Integrating symbolic reasoning with neural networks results in a large space of potential methods (see [Chaudhuri et al., 2021] for a comprehensive survey). Our MetaQNL/MetaInduce is more on the symbolic extreme within this space. Therefore, we propose future extensions incorporating neural components to achieve a more balanced integration that harnesses the strengths of both symbolic reasoning and neural networks. And a stronger soft matching mechanism is one promising way of implementing this integration.
>
>
> * [Garcez et al., 2020] Garcez, Artur d'Avila, and Luis C. Lamb. "Neurosymbolic AI: the 3rd wave." arXiv preprint arXiv:2012.05876 (2020).
> * [Gary, 2020] Marcus, Gary. "The next decade in ai: four steps towards robust artificial intelligence." arXiv preprint arXiv:2002.06177 (2020).
> * [Chaudhuri et al., 2021] Chaudhuri, Swarat, et al. "Neurosymbolic Programming." Foundations and Trends® in Programming Languages 7.3 (2021): 158-243.
>
>
> ## Low Rule Counts Because of Simple Datasets
>
> We agree with the reviewer that more complex datasets require a larger number of rules. This is also observed in our experiments. Our method learned 7 rules for MiniSCAN, 20 rules for SCAN, and ~100 rules for RuleTaker. However, our model size (#rules) is adaptive to the dataset’s complexity, which is a desirable property that neural networks do not have.
>
>
> ## Additional Literature on Reasoning in Natural Language
>
> Thanks you, and we will discuss them in the next revision.

---

### Review · Reviewer_kVrX · 2023-01-31

**Summary Of Contributions:**

The authors propose an approach for learning symbolic rules from data. This differs from the most prevalent types of learners in that what is learned is a symbolic rule (rather than weights in a neural network), and it differs from the best-known rule-based systems in that the rules are learned rather than hand-coded. The system includes both an abstract language for expressing rules called MetaQNL and a learning algorithm for learning rules called MetaInduce. The authors also augment these components with a soft matching mechanism that can enable the system to handle messy natural language inputs. These approaches provide very strong results on several synthetic reasoning benchmarks as well as respectable results on some more natural real-world datasets.

**Audience:**

Yes

**Claims And Evidence:**

Yes

**Requested Changes:**

Critical to securing my recommendation for acceptance:

- RC1: One branch of literature that is missing from the literature review is work in the tradition of Bayesian methods and probabilistic modeling. There have been prominent works in this tradition about learning symbolic rules from data, so it seems like this area should be discussed at least briefly. Here are some representative publications from this area:
    - Piantadosi et al: The Logical Primitives of Thought: Empirical Foundations for Compositional Cognitive Models
    - Goodman et al: A rational analysis of rule-based concept learning
    - Yang & Piantadosi: One model for the learning of language

- RC2: The figures are missing labels, so you can’t see which figure is “Figure 2”



Not critical to securing my recommendation:
- RC3: The first 2 paragraphs include many claims about the current state of machine learning. Many of these claims do not have citations (e.g., rule-based systems being brittle; neural nets requiring large amounts of training data and generalizing poorly). It would be nice to add citations for these points (but I labeled this “not critical” because I think all of these points represent standard views in the field, so probably no one will doubt these claims even without citations. Still, citations could be helpful for someone from a different field approaching this paper).
- RC4: I found it a little confusing how “r” and “cr” could refer to either a rule or a Boolean variable corresponding to the rule - maybe rename the boolean variable to “b_r” and “b_cr” or something?
- RC5: A fair number of examples from the datasets are ungrammatical e.g., “An elephants smiles”, “The elephant be not tall”). Is this just how the dataset is formatted? If so, it might be worth adding a footnote at the first instance saying “In this dataset, for simplicity, sentences do not perfectly follow English grammar.”
- RC6: Use of the term “state-of-the-art”: When it is said that a system achieves state-of-the-art results, I typically take that to mean it outperforms all previous systems. However, if I understand correctly, several of the cases where this paper uses the term “state-of-the-art” (e.g., SCAN) instead mean matching previous work, not exceeding it. So I would prefer it if the wording was clearer, perhaps saying “we match or exceed the previous state-of-the-art” so that it is clear that some of the results are matching rather than exceeding.
- RC7: “In principle, MetaQNL can solve any task to the extent that the task is solvable by computer programs.” To me, “solve” suggests “find a solution to.” However, MetaQNL doesn’t  find the solution, it just represents it. So it might be more accurate to say “MetaQNL can represent a solution for any task which has a solution that can be represented by a computer program.” (I admit that this sounds clunkier, however!)
- RC8: “our method learns interpretable morphological rules; e.g., the suffix áramos indicates the paste tense.“ I don’t speak Spanish - is this indeed a past-tense suffix in Spanish? Could be worth mentioning. (Also, note that there is a typo: “paste” instead of “past”).


**Strengths And Weaknesses:**

Strengths:

- S1: The approach is well-motivated through discussion of how rules and learning are not incompatible

- S2: The proposed meta-language and algorithm are both described clearly (with a couple minor exceptions - see below), and the design decisions seem sensible

- S3: The experiments are thorough, covering a range of domains

- S4: The results are impressive. Beyond the numerical results, what I found particularly exciting was some additional strengths of their approach compared to neural networks: First, it learns explicit rules which can be examined for correctness (and in many cases the learned rules match ground-truth ones), and the learning process is efficient in time and parameters (at least for the relatively simple cases considered here - as the authors note, the current implementation would struggle to scale up).


Weaknesses:

- W1: There are some minor issues with the framing and explanation, which I have listed below under “Requested changes.” However, I did not find there to be any major weaknesses.

---

> ### Author Response · Authors · 2023-02-18
> **Individual Response to Reviewer kVrX**
>
> Thank you for your valuable feedback! Below we address your questions and concerns.
>
> ## Learning Symbolic Rules in the Literature of Probabilistic Modeling and Bayesian Methods
>
> Thanks for providing the references. They are indeed relevant, and we will discuss them in the next revision.
>
>
> ## Missing Label for Fig. 2
>
> Thanks for pointing that out. Fig. 2 is on page 6. We will fix the label in the next revision.
>
>
> ## Typos, Wording, Variable Namring, and Other Minor Issues
>
> Thanks for your suggestions! We’ll incorporate them in the next revision.

---

### Review · Reviewer_HDXx · 2023-02-07

**Summary Of Contributions:**

This paper proposes a framework and algorithm for learning logical rules from (possibly noisy) sentences in the style of inductive logic programming (ILP). The authors correctly note that existing methods for ILP are brittle and unable to handle domains with softer reasoning and noisier, more ambiguous input, as is commonplace in natural language.

This work is an attempt to bridge the gap of promises of ILP: a system that can infer rules from possibly noisy data, while gracefully handling natural language. It consists of the following steps:

1. Define the problem: a set of provable and unprovable examples, where each example is a tuple of (A, g), where A is the assumptions and g is the goal. The goal is to learn the simplest possible model M with rules consisting of transformations of the assumptions in A -> g that proves all of the provable examples and cannot prove the unprovable examples.
2. Rule proving: for each example in the dataset, propose rules consistent with each example (But not necessarily correct)
3. Use standard forward/backward chaining to construct proofs given the proposed rules.
4. Generalize instances of concrete rules (e.g. "if an elephant is big, it is heavy") to rules with variables ("in X is big, X is heavy") via a process called anti-unification
5. Simplify the model as much as possible, via rule pruning, represented as a MAXSAT problem.

The key innovation over prior work seems to be the attempt at soft matching: the ability for MetaInduce to handle software natural language statements, rather than rigid logical systems. The apply this system only at test time: given a set of assumptions and a rule that transforemrs assumptions into conclusions, instead of rigidly depending on exact string matching, they can use soft string matching, or train a transformer LM, to propose a list of approximate conclusions with scalar matching scores.

The authors show that their proposed method solves synthetic logical datasets like SCAN and RuleTaker at 100% accuracy (being more efficient than nerual approaches) and is also able to handle fuzzy ambiguous datasets, as evaluated on a morphological induction dataset and a natural language version of RuleTaker. They show promising results in this domain, again compared to neural network baselines.

**Audience:**

Yes

**Broader Impact Concerns:**

None that I can see.

**Claims And Evidence:**

Yes

**Requested Changes:**

As in the section above, I think this paper is possibly fine as is (borderline) but would be much more solid if some more explanation of differences (and/or empirical comparisons if warranted) to existing fuzzy ILP systems. These are not crucial to securing my recommendation but I believe would make the paper stronger.

Minor typo page 12: paste -> past.

**Strengths And Weaknesses:**

# Strengths
- Interesting and important avenue to apply ILP-style systems to fuzzier/more ambiguous datasets.
- Good variety of both synthetic and less synthetic datasets observed. I personally don't take much stake in the synthetic reuslts, as it's somewhat unsurprising that MetaInduce (and indeed any ILP system) can 100% such tasks with rigid logical grammars. But the more naturalistic datasets are a good attempt.
- Interesting proposal to blend pretrained LMs with rule based systems, with some promising results here.

# Weaknesses
My biggest concerns with the paper have to do with clarity and relation to existing work. Unfortunately I am not super familiar with the literature on ILP and continuous approximations thereof, so I am not in a super strong position to judge the novelty or relation to other work.
- However, from my cursory knoweldge of ILP, my main concern with the paper is that either empirical comparisons to related attempts at making "soft" versions of ILP is missing from this paper. Either that, or greater discussion about the differences between this work and existing work. For example, I am familiar with $\partial \text{ILP}$, by Evans and Grefenstette, one of the early attempts at learning softer ILP systems via gradient descent. But such approaches are not examined in this paper—they are dismissed as learning rules restricted to "binary predicates". As Evans and Grefenstette note, however, restricting to binary predicates does not result in a loss of generality, since any logical system can be decomposed into more but simpler binary rules. I don't beliee this is a constraint that is "domain-dependent and difficult to get right". Moreover, it's ideal if authors explain how the tasks they explore indeed leverage n > 2-ary predicates to a greater degree than that handleable by $\partial \text{ILP}$. Could authors respond to these points?
- Moreover, going off of the author's related work section, I think a bit more context is need. Authors mention several existing weaknesses of the existing work, e.g. Unification Networks, and dismiss them as "less general", with the system not supporting multi-step reasoning. These may be true, but one question I have is whether the tasks explored in this paper are indeed unsolable by these existing systems. Specifically, do the tasks in this paper indeed dependent on certain properties (multi-step reasoning, variable binding to sentence fragments) that are only handlable by MetaQNL? Some qualitative evidence that this is the case (e.g. some examples of multi-step proofs or variable bindings) would be nice to have here. For example some tasks still look like one-step reasoning (SCAN/Morphological analysis) if I'm not mistaken, so some clarification would be good here.
- Soft matching only at test time seems like a good first step, but also a fairy limited attempt. Being able to do soft matching at test time doesn't seem to matter if you need to use soft matching at training time in order to learn the right model in the first place. It seemsl ike the success of using soft matching at test time must be explained by the fact that tasks are easy enough to learn w/o soft matching at training time. Could authors comment on this?

# Conclusion

Overall, it is clear that MetaInduce is in a fairly early stage, with limited (although promising) experiments and a somewhat unclear comparison to the rest of the ILP literature. Overall, it is clear that the paper will be of interest to many in TMLR's audience, and I appreciate the concerted effort towards a paradigm that is *not* deep learning nowadays, so I am leaning positive on the paper as is (though I don't feel strongly). I'd be more strongly in favor of acceptance if authors compared to more existing approaches for neurosymbolic learning, or clarify more why such comparisons are not warranted beyond what is already discussed in the related work.

---

> ### Author Response · Authors · 2023-02-18
> **Individual Response to Reviewer HDXx (1/4)**
>
> Thank you for your valuable feedback! Below we address your questions and concerns.
>
> ## Examples of Multi-Step Reasoning
>
> Multi-step reasoning is common in MiniSCAN/SCAN and RuleTaker. For MiniSCAN, the entire dataset and the learned rules are included in Appendix D. All testing examples require at least 2 steps of reasoning. For RuleTaker, Fig. 1 in the ProofWriter paper [Tafjord et al., 2021] provides an example of multi-step reasoning. And Table 2 in our paper reports performance on proofs of different numbers of reasoning steps.
>
>
> ## Limitation of Applying Soft Matching Only in Testing
>
> We agree with the reviewer that soft matching only in testing is a limitation of our current method. Applying soft matching in training can potentially enable MetaQNL/MetaInduce to work on more challenging datasets. However, it is out of the scope of this paper and left for future work.
>
>
> ## Empirical Comparisons with Existing Neurosymbolic Approaches
>
> Please see the common response at the top.
>
> ## Connections to Inductive Logic Programming (ILP)
>
> In our paper, “[Evans and Grefenstette, 2018] restricts to binary predicates” was intended to give examples of syntactic constraints in ILP approaches, not to claim it to be the main difference between our method and [Evans and Grefenstette, 2018]. We are sorry if this caused confusion and will revise the paper accordingly.
>
> This is our view of how MetaQNL distinguishes from ILP in handling natural language:  **ILP needs predicates manually defined by humans, as well as a highly accurate semantic parser**, both of which are extremely challenging (more on this point later). This also applies to [Evans and Grefenstette, 2018]. Their method is only applicable to tasks with predefined predicates. Due to this reason, we cannot compare with ILP empirically on benchmarks such as RuleTaker without manually defining the predicates.
>
> The comment that syntactic constraints in ILP are "domain-dependent and difficult to get right" is not specific to  [Evans and Grefenstette 2018] but generally applies to other forms of constraints, such as mode declaration and metarules. Similar comments are also expressed by existing surveys on ILP, e.g., Sec. 4.4 in [Cropper and Dumančić, 2022].
>
> Below we present a more detailed discussion of our method vs. ILP. We’ll incorporate elements from the discussion into the next revision.

---

> > ### Author Response · Authors · 2023-02-18
> > **Individual Response to Reviewer HDXx (2/4)**
> >
> > ### Differences in Representation Language
> >
> >
> > The most important difference is that we learn MetaQNL rules, whereas ILP learns rules in first-order logic programs, such as Prolog and Datalog. MetaQNL can express logic programs, but it can also express arbitrary natural language sentences. In contrast, logic programs cannot handle natural language easily. It needs to map sentences to literals in first-order logic. This further requires a predefined ontology of all objects and predicates, as well as a highly accurate semantic parser, both of which are infeasible.
> >
> >
> > For a concrete example, we have shown MetaQNL to achieve almost perfect accuracy on the RuleTaker dataset. We learn templated sentences such as `[X] [Y] things are [Z]` from raw texts. If we wanted to apply ILP to RuleTaker, we would need to manually design predicates such as `_-_-things-are-_(x, y, z)`. To really make ILP work for RuleTaker, we would need at least the following components:
> >
> >
> > * We first have to manually design all predicates such as `_-is-_(x, y)` and `_-_-things-are-_(x, y, z)` for representing assumptions and goals. Most ILP systems cannot invent any new predicate beyond those given by the user [Cropper and Dumančić, 2022]. Some ILP systems are capable of predicate invention, but the invented predicates are only used as intermediate results in reasoning, not for representing the assumptions and the goal.
> > * Then we need a perfect semantic parser for converting sentences to first-order logic. This is non-trivial even for synthetic datasets like RuleTaker, because we have no labeled data to train such a semantic parser via supervised learning.
> >
> >
> > All of the above are non-trivial and highly domain-specific. As surveyed by  [Cropper and Dumančić, 2022], ILP hasn't been shown to achieve state-of-the-art in domains that are non-trivial to formalize, such as (even synthetic) natural language. Therefore, we believe MetaQNL can solve tasks that are not easily solvable by ILP.

---

> > > ### Author Response · Authors · 2023-02-18
> > > **Individual Response to Reviewer HDXx (3/4)**
> > >
> > > ### Differences in the Algorithm for Learning Rules
> > >
> > >
> > > We also differ from ILP in the algorithm for learning rules. Encoding rule induction as a MAX-SAT problem, MetaInduce is inspired by existing *meta-level approaches* to ILP  [Cropper and Dumančić, 2022], but it is novel in the following aspects:
> > >
> > >
> > > **Constraints on the rule space**: Our approach for constraining the rule space is more general than ILP. ILP requires the user to supply very restricted rule templates, e.g., in the form of meta-rules. Taking the ILP system Metagol [Muggleton et al., 2015] as an example, in order to learn the logic program:
> > > ```
> > > grandparent(X, Z) :- parent(X, Y), parent(Y, Z)
> > > parent(X, Y) :- father(X, Y)
> > > parent(X, Y) :- mother(X, Y)
> > > ```
> > > , the user has to provide the following meta-rules as the template:
> > > ```
> > > P(X, Z) :- Q(X, Y), R(Y, Z)
> > > P(X, Y) :- Q(X, Y)
> > > P(X, Y) :- Q(X, Y), R(X, Y)
> > > ```
> > > , where `P`, `Q`, and `R` are higher-order variables that will be tied to concrete predicates by the learning algorithm. Suitable rule templates are critical for ILP to work well. However, as noted by  [Cropper and Dumančić, 2022], they are highly domain-specific and difficult to determine when the ground truth solution is unknown, which is a major obstacle to ILP. In contrast, we constrain the rule space using the rule proposer. It is also domain-specific but is easier to define and requires less domain knowledge from the user.
> > >
> > >
> > >
> > >
> > > **Search algorithm**: Similar to meta-level ILP approaches, we delegate rule induction to existing solvers (SAT, MAX-SAT, ASP, etc.). However, we differ in how to encode the problem, due to our unique representation language and hypothesis space. We illustrate these differences by comparing with the closest methods in ILP: ASPAL [Corapi et al., 2011], ProSynth [Raghothaman et al., 2019], ILASP [Law et al., 2014], Apperception [Evan et al. 2021], and Popper [Cropper and Morel, 2021]
> > >
> > >
> > > * Encoding rules vs. encoding proofs: Most meta-level ILP approaches (e.g., ASPAL, ILASP, Apperception, and Popper) directly encode candidate rules in answer set programming (ASP) and ask the ASP solver to find a subset of them, without a separate theorem proving stage. This is possible because they learn rules in Prolog, Datalog, or ASP, whose formal semantics can be encoded in ASP. In contrast, MetaQNL rules cannot be encoded in existing solvers directly. Therefore, we encode the proofs found by a prover.
> > > * Encoding proofs has also been explored in ProSynth for provenance-guided synthesis of Datalog programs. However, our MAX-SAT encoding extends the SAT encoding of ProSynth. First, we encode the disjunction of all proof paths (page 7), whereas only one proof path is available in ProSynth due to limitations of how provenance works. This is an important difference, because there are inevitably many proof paths if the rule proposer is sufficiently general. Second, we have additional constraints regarding rule instantiation (page 7). This is because MetaQNL rules form a hierarchy, whereas rules in ProSynth are independent of each other. The hierarchy is important for learning abstract rules from concrete ones. Third, we can tolerate noise, whereas ProSynth cannot. We use soft constraints to enforce the training examples to be provable (page 7), but ProSynth uses hard constraints. Allowing some examples to be unprovable is important for MetaInduce to learn compact and generalizable rules.

---

> > > > ### Author Response · Authors · 2023-02-18
> > > > **Individual Response to Reviewer HDXx (4/4)**
> > > >
> > > > * [Evans and Grefenstette, 2018] Evans, Richard, and Edward Grefenstette. "Learning explanatory rules from noisy data." Journal of Artificial Intelligence Research 61 (2018): 1-64.
> > > > * [Cropper and Dumančić, 2022] Cropper, Andrew, and Sebastijan Dumančić. "Inductive logic programming at 30: a new introduction." Journal of Artificial Intelligence Research 74 (2022): 765-850.
> > > > * [Muggleton et al., 2015] Muggleton, Stephen H., Dianhuan Lin, and Alireza Tamaddoni-Nezhad. "Meta-interpretive learning of higher-order dyadic datalog: Predicate invention revisited." Machine Learning 100 (2015): 49-73.
> > > > * [Corapi et al., 2011] Corapi, Domenico, Alessandra Russo, and Emil Lupu. "Inductive logic programming in answer set programming." International conference on inductive logic programming. Springer, Berlin, Heidelberg, 2011.
> > > > * [Raghothaman et al., 2019] Raghothaman, Mukund, et al. "Provenance-guided synthesis of Datalog programs." Proceedings of the ACM on Programming Languages 4.POPL (2019): 1-27.
> > > > * [Law et al., 2014] Law, Mark, Alessandra Russo, and Krysia Broda. "Inductive learning of answer set programs." European Workshop on Logics in Artificial Intelligence. Springer, Cham, 2014.
> > > > * [Evan et al. 2021] Evans, Richard, et al. "Making sense of sensory input." Artificial Intelligence 293 (2021): 103438.
> > > > * [Cropper and Morel, 2021] Cropper, Andrew, and Rolf Morel. "Learning programs by learning from failures." Machine Learning 110.4 (2021): 801-856.

---

### Author Response · Authors · 2023-02-18
**Common Response**

We thank all reviewers for their thoughtful comments! We are encouraged that they agree we address an important and exciting problem (HDXx, 5gc4). Reviewers also consider our approach novel, general (5gc4), well-motivated (kVrX), and our experimental results promising (HDXx), strong (kVrX), and comprehensive (5gc4). Upon the request of Reviewer HDXx and 5gc4, here we discuss comparisons with existing neurosymbolic approaches. Other questions and concerns are addressed in separate comments for each individual reviewer.

## HDXx, 5gc4: Empirical Comparisons with Existing Neurosymbolic Approaches

We only compared with neural approaches, as many existing neurosymbolic approaches are not readily applicable to our task setting without significant manual design. Below we explain individually why the neurosymbolic approaches in the related work section are not suitable for a direct empirical comparison with our method:
* [Rocktäschel and Riedel, 2017],  [Evans and Grefenstette, 2018], [Kathryn and Mazaitis, 2018] and [Saparov and Mitchell, 2022] require human-defined predicates.
* [Lee et al. 2016] and [Weber et al. 2019] require semantic parsing.
* Natural Logic ([McAllester and Givan, 1993] and [MacCartney and Manning, 2007]) can only handle monotonicity reasoning.
* [Grefenstette, 2013] focuses on theoretical discussions of how to tensors can be used to capture quantifier-free predicate calculus. It does not include a practical method that we can compare with.
* Methods performing “soft” reasoning in natural language with neural networks ([Clark et al., 2020], [Tafjord et al., 2021], [Yang et al., 2022], and papers suggested by Reviewer 5gc4): We have compared with [Tafjord et al., 2021]. Other methods are more structured, but they cannot outperform [Tafjord et al., 2021] on RuleTaker, as the accuracy of [Tafjord et al., 2021] is already close to 100%.
* ILP methods ([Muggleton, 1991], [Raghothaman et al., 2019]): Please see our response to Reviewer HDXx.


Furthermore, we have compared our method with state-of-the-art neural baselines for all benchmarks used in this paper (MiniSCAN, SCAN, RuleTaker, and SIGMORPHON). These baselines have achieved almost perfect accuracy. Therefore, including neurosymbolic approaches will not lead to stronger baselines in terms of accuracy.

---

### Decision · Action_Editors · 2023-04-24

**Recommendation:** Accept with minor revision

**Comment:**

This paper presents MetaQNL, a framework for learning logical rules from natural language data. An important component of the proposed method is the soft matching, removing thus some of the brittleness of previous work. The method is evaluated on 4 datasets reporting positive results wrt to comparisons.

All reviewers agree that this is a very interesting submission and of interest to the TMLR audience. However all reviewers have mentioned a number of discussions currently missing from the paper. Hence, I’m recommending acceptance with minor revisions.

Below are things that would be useful to get into the final manuscript.
* extend the discussion around scalability of method, per Reviewer 5gc4
* add additional literature on reasoning in NL and probabilistic modeling (per Reviewers kVrX and 5gc4)
* include discussion on why some comparisons are not made (per Reviewer HDXx)


**Audience:**

Yes

**Claims And Evidence:**

Yes